# ODNet: Opinion Dynamics-Inspired Neural Message Passing for Graphs and Hypergraphs

**Bingxin Zhou**                                                           *bingxin.zhou@sjtu.edu.cn*
*Institute of Natural Sciences*
*Shanghai Jiao Tong University*

**Outongyi Lv**                                                                *harry_lv@sjtu.edu.cn*
*Institute of Natural Sciences*
*Shanghai Jiao Tong University*

**Jing Wang**                                                          *zhenjiaofenjie@sjtu.edu.cn*
*School of Oceanography*
*Shanghai Jiao Tong University*

**Xiang Xiao**                                                             *zjxiao2018@sjtu.edu.cn*
*School of Life Sciences and Biotechnology*
*Shanghai Jiao Tong University*

**Weishu Zhao**                                                               *zwsh88@sjtu.edu.cn*
*School of Life Sciences and Biotechnology*
*Shanghai Jiao Tong University*

**Reviewed on OpenReview:** *https://openreview.net/forum?id=ytKFKoCpyK*

## Abstract

Neural message passing serves as a cornerstone framework in graph neural networks, providing a clear and intuitive mathematical guideline for the propagation and aggregation of information among interconnected nodes within graphs. Throughout this process, node representations undergo dynamic updates, considering both the individual states and connections of neighboring nodes. Concurrently, social networks, as prominent forms of interconnected data, form dynamic systems that achieve stability through continuous internal communications and opinion exchanges among social actors along their social ties. Drawing upon the shared concepts between these two domains, our study establishes an explicit connection between message passing and opinion dynamics in sociology. Moreover, we introduce a novel continuous message passing scheme termed ODNET, which integrates bounded confidence to refine the influence weight of local nodes for message propagation. By adjusting the similarity cutoffs of bounded confidence and influence weights within ODNET, we define opinion exchange rules that align with the characteristics of neural message passing and can effectively mitigate the oversmoothing issue. We extend the framework to hypergraphs and formulate corresponding continuous message passing rules, which reveal a close association with particle dynamics. Empirically, we showcase that ODNET enhances prediction performance across various social networks presented as homophilic graphs, heterophilic graphs, and hypergraphs. Notably, our proposed ODNET outperforms existing GNNs with its straightforward construction and robust theoretical foundation.

## 1 Introduction

Graph neural networks have emerged as a powerful deep learning framework for analyzing the intercommunication of entities on a graph, offering versatile tools for node, edge, and graph-level representation learning

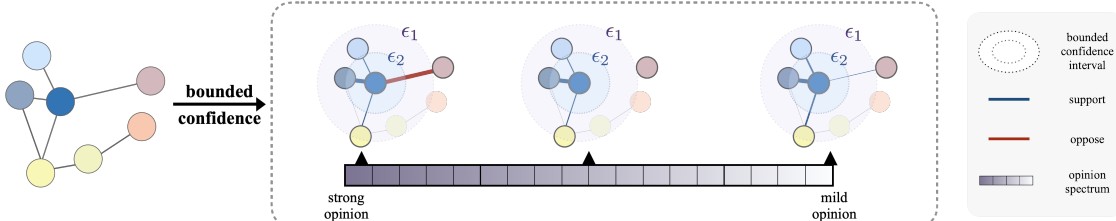

Figure 1: An illustration of the propagation rule in ODNet with bounded confidence. Colored nodes represent individuals with different opinions. The central node (dark blue) interacts with light blue nodes sharing similar opinions and a red node with significantly divergent thoughts. During message passing, ODNet enhances the aggregation weight of similar nodes (light blue) and adjusts the weights of conflict nodes (red). Bounded confidence defines the range of similar/conflict nodes and the extent of adjustments, and it is influenced by the graph's position on the opinion spectrum. See Section 5 for modeling details.

(Scarselli et al., 2008; Bronstein et al., 2017). The primary challenge in designing GNN models lies in effectively aggregating information based on local interactions for efficiently extracting hidden representations. This design philosophy has been generalized as *neural message passing* (Gilmer et al., 2017) and later became a fundamental feature extraction unit of graph for aggregating features of neighbors.

Concurrently, *opinion dynamics* studies the evolving social relations with the opinions dynamically changed over time through interactions (Acemoglu & Ozdaglar, 2011). A *social network* establishes a system with individual social actors and the social ties among them. It uses a graph as an abstract representation of a group's structure (Proskurnikov & Tempo, 2017). To analyze the formation, evolution, and spread of opinions within social networks or communities, statistical physics has contributed by introducing methods and tools from dynamical systems theory (Helbing, 2010; Weidlich, 2006).

Despite originating from different disciplines, neural message passing and opinion dynamics share core principles of information propagation and aggregation within interconnected systems. Both fields focus on understanding how local interactions among individuals (nodes) contribute to the emergence of global behaviors in social networks (graphs). This inherent connection motivates our exploration of an efficient alternative to neural message passing inspired by popular models in opinion dynamics. We aim to address three key research questions: (1) *How to integrate models of opinion dynamics into the neural message passing framework?* (2) *What are the implications of opinion dynamics models' properties for mitigating the oversmoothing issue in neural message passing?* (3) *How can we apply opinion dynamics-inspired neural message passing rules to typical graphs encountered in social networks, such as heterophilic graphs and hypergraphs?*

To address the first two questions, we analyze the formulation and dynamic behavior of two representative models in opinion dynamics: the French-DeGroot (FD) model (DeGroot, 1974; French Jr, 1956) and the Hegselmann-Krause (HK) model (Rainer & Krause, 2002). We reformulate the FD and HK models to establish explicit connections with the message passing framework. Based on the unified formulation, we analyze the convergence behavior of the two models and provide a novel explanation of the oversmoothing issue in GNNs from the perspective of opinion dynamics. We demonstrate that the FD message passing converges exponentially to a stable state when exhibiting strong local connectivities. This property is frequently observed in hypergraphs and significantly contributes to the oversmoothing issue. Alternatively, the HK model mitigates the issue by introducing heterophilious dynamics to avoid consensus or environmental averaging.

Based on this connection, we answer the third question by integrating the concept of *bounded confidence* from the HK model and introduce an opinion dynamics-inspired message passing formulation named ODNet. A few recent studies combine opinion dynamics to define propagation rules on graphs. However, they are based on position-based first-order message passing (Okawa & Iwata, 2022) or ignore data-driven propagation rules (Giráldez-Cru et al., 2022), resulting in limited expressivity. This framework, instead, incorporates a confidence-based filtration mechanism on initial edge connections and incorporates second-order diffusion processes. By considering the similarity of node pairs and connection proximity, ODNet aggregates neighboring information through automatic adjustment to edge weights. Through its piecewise message passing schemes, the model can strengthen, weaken, or remove initial links within the graph. Furthermore, it al-

lows for the assignment of negative weights to capture conflicting viewpoints from neighboring nodes with substantial disparities. This capability proves essential especially when examining heterophilic networks, where connected entities exhibit dissimilar characteristics. The propagation scheme mirrors the dynamic spreading of opinions within a community, where effective communication channels converge individuals or agents towards consensus or a few dominant viewpoints (Figure 1). When exchanging information, individuals tend to support opinions aligned with themselves. However, when encountering significantly divergent thoughts, individuals may choose to disregard or oppose them, depending on position of the social network on the opinion spectrum. For example, researchers (on the right side of the spectrum) typically concentrate on studies within their expertise but are open to learning new perspectives from other domains, whereas politicians (on the left side of the spectrum) often encounter strong conflicts and resist propositions from competing parties.

We evaluate the versatile ODNET across three categories of graphs: homophilic graphs, heterophilic graphs, and hypergraphs, each characterized by unique properties resembling real-world social networks such as the small-world effect (Girvan & Newman, 2002) and scale-free networks (Barabasi & Oltvai, 2004). Empirically, the introduced piecewise aggregation behavior enhances the performance of well-established message passing methods. In summary, our work has three main contributions: (1) **We demonstrate an explicit connection between opinion dynamics and the neural message passing framework**, which used to be two parallel research fields in modeling social networks (Section 4). By formulating opinion dynamics models under neural message passing frameworks, we offer a new interpretation of oversmoothing from the perspective of opinion dynamics. (2) **We propose ODNet, a novel message passing scheme inspired by opinion dynamics** (Section 5.1). ODNET provides a promising alternative to traditional neural message passing methods. It incorporates bounded confidence to refine the influence weight of local node features, boasting a clean formulation, solid theoretical support, and effective mitigation of the oversmoothing issue. Notably, the introduction of bounded confidence does not introduce additional computational cost to the propagation. (3) **We defines the continuous ODNet for hypergraphs and provide theoretical analysis** (Section 5.2). By extensive experiments, we validate its superior performance on different types of graphs and compare it with both classic and SOTA discrete or continuous message passing schemes.

## 2 Related Work

**Neural Message Passing on Graphs and Hypergraphs** Neural message passing establishes a general computational rule for updating node representations in attributed graphs (Gilmer et al., 2017). By implementing different types of propagation rules, information at the node and edge levels is communicated among neighborhoods. In parallel, Feng et al. (2019) and Gao et al. (2022b) established a general convolution framework employing the incidence matrix for hypergraph learning. Various techniques have also undergone expansion, such as the attention mechanism (Bai et al., 2021) and spectral theory (Yadati et al., 2019).

**Continuous Message Passing and the Oversmoothing Issue** The concept of continuous GNN dynamics was first introduced by Poli et al. (2020) and Chamberlain et al. (2021), which explicitly generalizes discrete propagation frameworks such as GAT (Veličković et al., 2018). This diffusion framework has seen active extensions into more continuous graph convolutions with additional considerations of graph rewiring (Brandstetter et al., 2021; Bodnar et al., 2022) Meanwhile, the energy conservation of second-order diffusion processes circumvents the oversmoothing issue and training instability by capturing long-range interactions (Wang et al., 2023; Han et al., 2024).

**Opinion Dynamics for Social Networks and Graph Neural Networks** The natural connection between graphs and social networks has inspired pioneering works exploring the possibility of extending the analysis of social networks to graph neural networks. Sheaf networks build connections with socialdynamics by separating messages from nodes into public and private opinions (Caralt et al.; Duta et al., 2024; Zaghen et al., 2024; Hansen & Ghrist, 2021). SINN (Okawa & Iwata, 2022) follows a first-order formulation of information exchange to capture the positions of individuals without considering mechanisms to prevent excessive attraction or repulsion from neighbors, thus ignoring the inherent oversmoothing issue.To extend to the second-order formulation with Newtonian dynamics, social networks analysis defines *collective dynamics*

(Carrillo et al., 2010; Motsch & Tadmor, 2014) that depicts the phenomenon of agents moving together based on attraction and repulsion forces (Cucker & Smale, 2007; Fang et al., 2019; Jin & Shu, 2021b). ATBCR (Giráldez-Cru et al., 2022) followed opinion dynamic rules and introduced the concept of attraction and repulsion between nodes. However, the explicit propagation rule excludes learnable message passing layers, which restricts its performance on more sophisticated networks such as heterophilic graphs.

## 3 Graphs from the View of Message Passing and Opinion Dynamics

### 3.1 Graphs and Hypergraphs

A graph $\mathcal{G}[\boldsymbol{W}] = (\mathcal{V}, \mathcal{E}[\boldsymbol{W}])$ of $N$ nodes can be associated with any square non-negative weight matrix $\boldsymbol{W} \in \mathbb{R}^{N \times N}$, where $\mathcal{V}$ represents the node set and $\mathcal{E}$ is the edge set. We denote $x_i$ as the feature of node $v_i$ or the opinion of individual $i$. $\mathcal{G}$ is *strongly connected* (Godsil & Royle, 2001) if there exists a path from every node to every other node. A cycle is a directed path that both begins and ends at the same node with no repeated nodes except for the initial/final one. The length of a cycle is defined by the number of edges in the cyclic path. The periodicity of a graph is defined as the smallest integer $k$ that divides the length of every cycle in the graph. When $k = 1$, $\mathcal{G}$ is termed *aperiodic* (Bullo et al., 2009).

A hypergraph is a generalization of a graph in which an edge can connect any number of vertices. It can be denoted by a triple $\mathcal{H}[\boldsymbol{W}^h] = \{\mathcal{V}, \mathcal{E}, \boldsymbol{W}^h\}$. To avoid notation confusion, we still use $\mathcal{V}$ for the set of nodes and $\mathcal{E}$ for the set of hyperedges. We set $|\mathcal{V}| = N$ and $|\mathcal{E}| = M$, and $\mathcal{E}(i)$ denotes a set containing all the nodes sharing at least one hyperedge with node $i$. Usually, $\boldsymbol{W}^h$ is a diagonal matrix for hyperedges, where $W_{ee}^h$ represents the weight of the hyperedge $e$. In this paper, we extend the weight representation to a triple tensor $\boldsymbol{W}^h \in \mathbb{R}^{N \times N \times M}$, where $w_{i,j}^e$ designates an element in $\boldsymbol{W}^h$. The incidence matrix $\boldsymbol{H} \in \mathbb{R}^{N \times M}$ defines $H_{i,e} = 1$ if the node $i$ belongs to the hyperedge $e$, otherwise $H_{i,e} = 0$. We generalize the indicator within $\boldsymbol{W}^h$ by setting $w_{i,j}^e$ as nonzero if node $i, j$ are connected by a hyperedge $e$, otherwise 0.

### 3.2 Neural Message Passing on Graphs and Hypergraphs

Neural Message Passing (Gilmer et al., 2017) stands as the prevailing propagator for updating node representations in GNNs. We denote $\boldsymbol{x}_i^{(k-1)}$ as the features of node $i$ in layer $(k-1)$ and $a_{j,i} \in \mathbb{R}^d$ as the edge features from node $j$ to node $i$. A message passing layer reads as follows:

$$\mathbf{x}_i^{(k)} = \gamma^{(k)} \left( \mathbf{x}_i^{(k-1)}, \square_{j \in \mathcal{N}_i} \, \phi^{(k)} \left( \mathbf{x}_i^{(k-1)}, \mathbf{x}_j^{(k-1)}, a_{j,i} \right) \right), \tag{1}$$

where $\square$ denotes a differentiable, node permutation-invariant function, such as summation, mean, or maximization. The $\gamma$ and $\phi$ denote differentiable functions such as MLPs (Multi-Layer Perceptrons), and $\mathcal{N}_i$ represents the set of one-hop neighbors of node $i$. The message passing mechanism updates the feature of each node by aggregating their self-features with neighbors' features. On hypergraphs, the message passing scheme considers interactions among multiple nodes reflected in a hyperedge. At the $k$th layer:

$$\mathbf{x}_i^{(k+1)} = \Psi^{(k)} \left( \mathbf{x}_i^{(k)}, \Phi_{1, e \in \mathcal{E}(i)} \left( e, \Phi_{2, j \in e}^{(k)}(\{\mathbf{x}_j^{(k)}\}, \{a_{j,i}^e\}) \right) \right), \tag{2}$$

where $\Phi_1^{(k)}(\cdot)$ denotes a differentiable, hyperedge permutation-invariant function, and $\Phi_2^{(k)}(\cdot)$ is a differentiable, node permutation invariant function. The $\Psi^{(k)}(\cdot)$ denotes another differentiable function of propagation, and $j \in e$ implies $H_{j,e} = 1$ or $a_{j,i}^e \neq 0$.

## 4 Opinion Dynamics

This section introduces the FD and HK models, two iconic models in opinion dynamics, following a discussion on the interconnection of their formulation with message passing and the oversmoothing issue in GNNs.

### 4.1 French-DeGroot Model

The *French-DeGroot* (FD) model (DeGroot, 1974) is a groundbreaking agent-based model that simulates the evolution of opinions. In a given population of $N$ individuals, each individual holds an opinion $\boldsymbol{x}_i(k)$ at discrete time instances $k = 0, 1, \cdots$. An individual's opinion is evolved by

$$\boldsymbol{x}_i(k+1) = \sum_{j=1}^{N} w_{ij} \boldsymbol{x}_j(k), \tag{3}$$

where the non-negative *influence weight* $w_{ij}$ satisfies $\sum_{j=1}^{N} w_{ij} = 1$. If $w_{ij} > 0$, individuals $i$ and $j$ are considered neighbors. The influence weight signifies the relative impact that individual $j$ exerts on $i$ during each opinion update. Importantly, all individuals concurrently update their opinions at each time step. The FD model captures how individual opinions converge within a group that potentially leading to a consensus. This opinion pooling process could also be interpreted as a message passing scheme, which emulates how information is exchanged within a specific type of neural network.

**Connection to Neural Message Passing**   The FD model is often regarded as a micro-level model based on individuals simulating the evolution of individual opinions. We show that it shares similarities with GRAND (Chamberlain et al., 2021), which describes a diffusion process on graphs by connecting heat conduction with neural message passing. The connection between the two models is established through the discretization of a partial differential equation on graphs:

$$\frac{\partial}{\partial t} \boldsymbol{x}(t) = (\boldsymbol{A}(\boldsymbol{x}(t)) - \boldsymbol{I}_N) \boldsymbol{x}(t), \tag{4}$$

where $\boldsymbol{A}(\boldsymbol{x}(t))$ denotes the $N \times N$ attention matrix on nodes and $\boldsymbol{I}_N$ is an identity matrix. GRAND coincides with the FD model when $(\boldsymbol{A}(\boldsymbol{x}(t)) - \boldsymbol{I}_N)$ satisfies the row-stochastic property and a simple forward-Euler method is applied with a time step of one. This intriguing parallel between the two models highlights the interconnection of designs from the two different domains of social network modeling.

**Opinion Consensus and the Oversmoothing Issue**   A fundamental result regarding the convergence of the FD model is well-established, demonstrating that consensus is achieved exponentially fast for a strongly connected and aperiodic graph (Bullo et al., 2009; Proskurnikov & Tempo, 2017; Ye, 2019).

**Proposition 1.** *Consider the evolution of opinions $\boldsymbol{x}_i(k)$ for each individual $i$ within the network $\mathcal{G}[\mathbf{W}]$ according to (3). Assuming that $\mathcal{G}[\mathbf{W}]$ is strongly connected and aperiodic, and that $\mathbf{W}$ is row-stochastic. Define $\zeta$ as the dominant left eigenvector of $\mathbf{W}$, then $\lim_{k \to 0} \mathbf{x}(k) = (\zeta^\top \mathbf{x}(0)) \mathbf{1}_N$ exponentially fast.*

The proposition above shows that any graph with a self-loop is considered aperiodic, implying that exponential decay often occurs in graphs with strong connectivity. In GNNs, this phenomenon is known as *oversmoothing* (Nt & Maehara, 2019; Oono & Suzuki, 2019), which has been widely studied for its association with the exponential decay of the *Dirichlet energy* – a measure of feature convergence weighted by graph structure. Despite their distinct origins, these phenomena describe similar processes.

### 4.2 Hegselmann-Krause Model

In the FD model, each agent could interact with any other agent, regardless of their opinions. However, the previous analysis shows that such trivial aggregation behavior inevitably leads to oversmoothing the signals in the system. In real-life scenarios, individuals typically engage in conversations primarily with those who share similar viewpoints. In sociodynamics, this phenomenon is termed as collective behaviors, where a group of individuals collectively exhibit coordinated patterns. The *Hegselmann-Krause* (HK) model (Holm & Putkaradze, 2006; Kolokolnikov et al., 2013; Motsch & Tadmor, 2014) characterizes this behavior by

$$\boldsymbol{x}_i(k+1) = |\mathbb{B}(i, \boldsymbol{x}_i))|^{-1} \sum_{j \in \mathbb{B}(i, \boldsymbol{x}_i)} \boldsymbol{x}_j(k), \tag{5}$$

where the *bounded confidence* $\mathbb{B}(i, \boldsymbol{x}_i) = \{j : |\boldsymbol{x}_j(k) - \boldsymbol{x}_i(k)| < \epsilon\}$ includes all the peers $j$ associated with the individual $i$ whose opinions diverges within a confined region $\epsilon_i \in \mathbb{R}$. This parameter defines the degree of

uncertainty or tolerance for opinion exchanges. It restricts that an individual $i$ with an opinion $\boldsymbol{x}_i$ will only engage in interactions with those of their peers whose opinions fall within a confined region.

**Consensus versus Opinion Polarization** The HK model is a first-order opinion dynamics model that directly incorporates the influence of neighboring opinions on individual updates through bounded confidence. Nevertheless, bounded confidence does not inherently prevent the oversmoothing issue. In scenarios of significant heterophily, agents preferentially connect with dissimilar others. In this case, the model often leads to consensus and mirrors the oversmoothing problem in GNNs. It thus becomes pivotal to guide the HK model to capture opinion polarization and form multiple distinct and stable clusters. One way to approach this target involves employing strategies like bi-clustering with repulsion mechanisms (Fang et al., 2019; Jin & Shu, 2021b; Wang et al., 2023), which ensures a lower bound for the Dirichlet energy of the input signal.

## 5 Opinion Dynamics-Inspired Message Passing

This section defines ODNET, a novel message passing framework that employs the *influence function* $\phi(s)$ with bounded confidences. We offer a comprehensive interpretation of each component in ODNET. We introduce the discrete and continuous forms of the proposed model on both graphs and hypergraphs.

### 5.1 ODNet on Graphs

**Discrete Formulation** Denote $\phi(\cdot)$ as a non-decreasing function of the similarity measure $s_{i,j}$ to node $i$ and node $j$. The update rule for ODNET reads:

$$\boldsymbol{x}_i(t+1) = \sum_{i=1}^{N} \phi(s_{ij})(\boldsymbol{x}_j(t) - \boldsymbol{x}_i(t)) + \boldsymbol{x}_i(t) + u(\boldsymbol{x}_i(t)). \tag{6}$$

The matrix $s_{ij}$ has different definitions, such as the normalized adjacency matrix (Kipf & Welling, 2017) and attention coefficients (Veličković et al., 2018). The control term $u(\boldsymbol{x}_i)$ is deployed for stability. One straightforward choice of formulation is $u(x_i) = \nabla P(x)|_{x=x_i}$ with a potential function $P(x)$, such that $\nabla P(x) \to \infty$ as $x \to \infty$ (Kolokolnikov et al., 2011). It acts as a moral constraint preventing individuals from resorting to extreme violence in conflict situations in sociodynamics.

**Continuous Formulation** In opinion dynamics, individual perspectives tend to evolve gradually rather than through sudden changes. This observation suggests enhancing the discrete message passing model into a continuous framework. This adaptation can be understood as transforming a traditional message passing approach into a numerical approximation of the continuous model by rewriting (4) into:

$$\frac{\partial \boldsymbol{x}_i(t)}{\partial t} = \sum_{i=1}^{N} \phi(s_{ij})(\boldsymbol{x}_j(t) - \boldsymbol{x}_i(t)) + u(\boldsymbol{x}_i). \tag{7}$$

The continuous formulation (7) facilitates the application of numerous numerical approximation methods, transforming it back into a discrete model equipped with a specific scheme for residual adjustment. This adaptability allows applying diverse Ordinary Differential Equation (ODE) solvers to ODNET, including Neural ODEs (Chen et al., 2018). The associated explorations can be found in Section 6.

**Choice of Influence Function** The monotonic influence function $\phi(\cdot)$ characterizes the influence weight associated with the similarity of node pairs. It describes the diverse behaviors of opinion exchange in a system, which is determined by the systems' positions along the opinion spectrum (Figure 1). We define a piecewise $\phi(\cdot)$ to delineate influence regions akin to bounded confidence. For instance, with

$$\phi(s) = \begin{cases} \mu s, & \text{if } \epsilon_2 < s \\ s, & \text{if } \epsilon_1 \leq s \leq \epsilon_2 \\ 0, & \text{otherwise,} \end{cases} \tag{8}$$

the propagation rule (6) can be rewritten as

$$\boldsymbol{x}_i(t+1) = \mu \sum_{\epsilon_2 < s_{i,j}} s_{i,j}(\boldsymbol{x}_j(t) - \boldsymbol{x}_i(t)) + \sum_{\epsilon_1 \leq s_{i,j} \leq \epsilon_2} s_{i,j}(\boldsymbol{x}_j(t) - \boldsymbol{x}_i(t)) + \boldsymbol{x}_i(t). \tag{9}$$

This formulation amplifies the influence weight for highly similar node pairs with coefficient $\mu > 0$ while cutting connections for node pairs with low similarity, resembling how individuals tend to ignore opinions beyond their bounded confidence.

The previous choice (8) corresponds to when individuals maintain neutral opinions toward each other. Conversely, when individuals holding significantly divergent opinions and the network leans towards the 'strong opinion' side of the spectrum, interactions may become hostile. In such cases, $\phi(\cdot)$ can be defined as

$$\phi(s) = \begin{cases} \mu s, & \text{if } \epsilon_2 < s \\ s, & \text{if } \epsilon_1 \leq s \leq \epsilon_2 \\ \nu(1-s), & \text{otherwise}, \end{cases} \tag{10}$$

where $\mu > 0$ and $\nu < 0$. This configuration allows the neural network to learn from *positive* neighbors with high similarities and to extract *negative* information from nodes with discrepancies. Specifically, the negative coefficient $\nu$ indicates that certain node pairs consistently repel each other, potentially leading to undesirable system expansion. Consequently, the previously introduced control term $u(x_i)$ is essential for the model towards enhanced system stability.

**Remark 1.** *The choice of $\phi(\cdot)$ can be guided by the graph's intrinsic characteristics or by using the* homophily level *(Zhou et al., 2023) as a quantitative measure. We recommend using (8) for homophilic graphs and (10) for heterophilic graphs. The empirical analysis is in Section 6.*

## 5.2 ODNet on Hypergraphs

It is conceivable to extend a graph diffusion framework, *e.g.*, GRAND, to hypergraphs with macroscopic interpretations. However, these diffusion-type dynamics at the macro level are susceptible to oversmoothing of feature evolution, similar to traditional GNNs. Especially on hypergraphs, since are generally characterized with denser local connections than traditional graphs, addressing the oversmoothing issue becomes particularly important when constructing message passing propagation rules for hypergraphs. In this section, we will first analyze the oversmoothing issue of diffusion-based hypergraph message passing, then propose the continuous update rule of ODNET for hypergraphs that circumvent the oversmoothing issue.

**Diffusion-based Hypergraph Message Passing** When incorporated with macroscopic interpretations, diffusion-based hypergraph message passing propagates the attractions of individuals or agents. Consider the node feature space $\Omega = \mathbb{R}^d$ and the tangent vector field space $T\Omega = \mathbb{R}^d$. For $\mathbf{x}, \mathbf{y} \in \Omega$ and $\mathfrak{x}, \mathfrak{y} \in T\Omega$ where $\mathfrak{x}_{i,j} = -\mathfrak{x}_{j,i}$, we adopt the following inner products:

$$\langle \mathbf{x}, \mathbf{y} \rangle = \sum_{i,j} \mathbf{x}_i \mathbf{y}_j, \qquad [\mathfrak{x}, \mathfrak{y}] = \sum_{i>j} \sum_{e \in \mathcal{E}} h_{i,j}^e \, \mathfrak{x}_{i,j} \mathfrak{y}_{i,j}. \tag{11}$$

Here $h_{i,j}^e$ represents a tuple related to node $i, j$ and the associated hyperedge $e$, where $h_{i,j}^e = 0$ if $H_{i,e} = 0$. We set $h_{i,j}^e$ to satisfy $\sum_{j} \sum_{e \in \mathcal{E}} h_{i,j}^e = 1$. For any $\mathfrak{u} \in T\Omega$, by the adjoint relation $[\mathfrak{u}, \nabla \mathbf{x}] = \langle \mathbf{x}, \operatorname{div} \mathfrak{u} \rangle$ with $\nabla \mathbf{x} = \mathbf{x}_j - \mathbf{x}_i$, we derive:

$$(\operatorname{div} \mathfrak{u})_j = \sum_{i} \sum_{e \in \mathcal{E}} h_{i,j}^e u_{i,j}. $$

This leads to a formal diffusion process of a hypergraph:

$$\frac{d\mathbf{x}_i}{dt} = \operatorname{div} \nabla \mathbf{x}_i = \sum_{j} \sum_{e \in \mathcal{E}} h_{i,j}^e (\mathbf{x}_j - \mathbf{x}_i). \tag{12}$$

For simplicity, we define the hypergraph operator $\mathcal{L} = I - (\sum_{e \in \mathcal{E}} h_{i,j}^e)$ and rewrite (12) as

$$\frac{d\mathbf{x}}{dt} = -\mathcal{L}\mathbf{x}.$$

When $\mathcal{L}$ is semi-positive definite (s.p.d), we define (12) as a diffusion-type propagation of a hypergraph. The different choices of $h_{i,j}^e$ lead to diverse diffusion-type equations. For example, when we take forward Euler discretization on (12) and define

$$\sum_{e \in \mathcal{E}} h_{i,j}^e = D_v^{-\frac{1}{2}} H W D_e^{-1} H^T D_v^{-\frac{1}{2}},$$

we have a simplified HGNN (Gao et al., 2022a) without channel mixing.

**Oversmoothing Problem of Hypergraph Message Passing**   To understand this claim, we first define the Dirichlet energy of a hypergraph $\mathcal{H}$ of vector field $\mathbf{x} \in \mathbb{R}^{N \times d}$ for diffusion-type hypergraph networks:

$$\mathbf{E}(\mathbf{x}) := \sum_{i,j=1}^{N} \sum_{e \in \mathcal{E}} H_{i,e} H_{j,e} \|\mathbf{x}_i - \mathbf{x}_j\|^2. \tag{13}$$

When formulating the Dirichlet energy on hypergraph, rather than following the conventional formulation of $\mathbf{E}(\mathbf{x}) := \text{tr}(\mathbf{x}^\top \mathcal{L} \mathbf{x})$, we adopt a simplified definition by Rusch et al. (2022) due to the non-deterministic nature of $\mathcal{L}$, which also facilitates a straightforward analysis of the variations in node features.

**Definition 1.** *Let $\mathbf{x}^l$ represent the hidden features at the $l^{th}$ layer. Oversmoothing in a hypergraph neural network is defined as the exponential decay of layer-wise Dirichlet energy as a function of layer depth $l$, i.e.,*

$$\mathbf{E}(\mathbf{x}^l) \leq C_1 e^{-C_2 l}, \tag{14}$$

*where $C_1$ and $C_2$ are positive constants.*

Given that $|\mathbf{x}| \leq C_1 e^{-\gamma t}$ with $\gamma$ being the smallest positive eigenvalue of $\mathcal{L}$, it is easy to see that oversmoothing is a common issue in general hypergraph diffusion networks. This observation comes from the diffusion structure, where the node features $\mathbf{x}$ decay exponentially to zero with the s.p.d kernel $\mathcal{L}$.

**Continuous Formulation of ODNet on Hypergraphs**   ODNET offers a comprehensive message passing framework that can be directly extended to hypergraphs. The primary distinction between hypergraphs over graphs lies in how hyperedges facilitate connectivity beyond traditional pairwise edges. Following the continuous representation of (12), we define:

$$\frac{\partial \boldsymbol{x}_i}{\partial t} = \sum_{e:i \in e} \sum_{j \in e} \phi(s_{i,j}^e)(\boldsymbol{x}_j - \boldsymbol{x}_i) + u(\boldsymbol{x}_i). \tag{15}$$

This model formulates collective behaviors-based diffusion-type dynamics for hypergraphs. It suggests that in a large community, information spreads through subgroups (nodes connected by a hyperedge) rather than solely through direct pairwise exchanges. As with graphs, the selection of $s_{i,j}^e$ can include different metrics, such as attention coefficients (Bai et al., 2021) or convolution coefficients (Gao et al., 2022a).

The HK model can be interpreted as a diffusion process on graphs featuring piecewise attraction and repulsion behaviors, as outlined in (7) and (15). When devising message passing aggregation rules, employing a comprehensive framework for microscopic models based on collective behaviors within a complete interaction system proves effective in mitigating the oversmoothing issue. The complete proof is in Appendix A.

**Remark 2.** *The diffusion process of a hypergraph by (12) is also closely related to self-organized dynamics in particle systems (Motsch & Tadmor, 2014), where (12) represents a particle dynamics scenario with $h_{i,j}^e$ signifying the interactive force between nodes $i, j$ under a specific field $e$. Note that (12) corresponds to a particular case of (15) where only attractive forces influence the message evolution. However, this assumption is not universal for particle systems, and as demonstrated above, it can lead to the oversmoothing issue.*

Table 1: Average test accuracy on **homophilic** graphs over 10 random splits.

| | **Cora** | **CiteSeer** | **PubMed** | **Coauthor CS** | **Computer** | **Photo** | **ogb-arXiv** |
|---|---|---|---|---|---|---|---|
| homophily level | 0.83 | 0.71 | 0.79 | 0.80 | 0.77 | 0.83 | 0.61 |
| GCN | 81.5±1.3 | 71.9±1.9 | 77.8±2.9 | 91.1±0.5 | 82.6±2.4 | 91.2±1.2 | 72.2±0.3 |
| MoNet | 81.3±1.3 | 71.2±2.0 | 78.6±2.3 | 90.8±0.6 | 83.5±2.2 | 91.2±2.3 | N/A |
| GraphSage-avg | 79.2±7.7 | 71.6±1.9 | 77.4±2.2 | 91.3±2.8 | 82.4±1.8 | 91.4±1.3 | N/A |
| GraphSage-max | 76.6±1.9 | 67.5±2.3 | 76.1±2.3 | 85.0±1.1 | N/A | 90.4±1.3 | N/A |
| GAT | 81.8±1.3 | 71.4±1.9 | 78.7±2.3 | 90.5±0.6 | 78.0±1.9 | 85.7±2.0 | 73.6±0.1 |
| GAT-PPR | 81.6±0.3 | 68.5±0.2 | 76.7±0.3 | 91.3±0.1 | **85.4±0.3** | 90.9±0.3 | N/A |
| CGNN | 81.4±1.6 | 66.9±1.8 | 66.6±4.4 | 92.3±0.2 | 80.3±2.0 | 91.4±1.5 | 58.7±2.5 |
| GDE | 78.7±2.2 | 71.8±1.1 | 73.9±3.7 | 91.6±0.1 | 82.9±0.6 | 92.4±2.0 | 56.7±10.9 |
| GRAND-l | 83.6±1.0 | 73.4±0.5 | 78.8±1.7 | 92.9±0.4 | 83.7±1.2 | 92.3±0.9 | 71.8±0.2 |
| SINN | 83.2±0.4 | 73.9±0.6 | 80.0±1.7 | 93.0±0.5 | 84.6±1.1 | 90.4 ±2.4 | N/A |
| ODNet (ours) | **85.7±0.3** | **75.5±1.2** | **80.6±1.1** | **93.1±0.7** | 83.9±1.5 | **92.7±0.6** | **74.65±0.16** |

## 6 Empirical Analysis

We validate the performance of ODNet through classic node-level representation learning tasks on a variety of social networks described by homophilic/heterophilic graphs or hypergraphs.

### 6.1 Experimental Protocol

**Training Setup** We compare our model to a diverse set of top-performing baseline GNN models, including classic message passings with discrete or continuous updating schemes, as well as the latest hypergraph models. For ODNet, we trained the model using a neural ODE solver with Dormand–Prince adaptive step size scheme (DOPRI5). For homophilic datasets, we utilized 10 random weight initializations and random splits, with each combination randomly selecting 20 instances from each class. In heterophilic and hypergraph datasets, we used the fixed 10 training/validation splits in Pei et al. (2020) and Yadati et al. (2019). All implementations are programmed with `PyTorch-Geometric` (version 2.0.1) (Fey & Lenssen, 2019) and `PyTorch` (version 1.7.0) and run on NVIDIA® Tesla A100 GPU with 6,912 CUDA cores and 80GB HBM2 mounted on an HPC cluster. All the details to reproduce our results have been included in the submission. The program will be publicly available upon acceptance. The searching space of hypergraphs is reported in Table 7. We compare ODNet with a variety of previous methods on graphs and hypergraphs (see Appendix B.1).

### 6.2 Node Classification on Graphs

**Benchmark Datasets** To evaluate a diverse set of scenarios, we encompass seven homophilic graphs (**Cora** (McCallum et al., 2000), **Citeseer** (Sen et al., 2008), **Pubmed** (Namata et al., 2012), **Coauthor CS** (Shchur et al., 2018), **Computer** (Namata et al., 2012), **Photo** (Namata et al., 2012), and **ogb-arXiv** Hu et al. (2020)), three heterophilic graphs (**Texas**, **Wisconsin**, and **Cornell** from the WebKB dataset (García-Plaza et al., 2016)). An overview of the statistical information for the six homophilic graphs and three heterophilic graphs is provided in Table 4 of Appendix B. The benchmarks used for evaluation cover a wide range of *homophily level* (Zhou et al., 2023). A low homophily level indicates that the dataset leans more towards being heterophilic, where most neighbors do not share the same class as the central node. Conversely, a high homophily level signifies that the dataset tends towards homophily, with similar nodes more likely to be interconnected.

**Result Analysis** Tables 1-2 present the average accuracy for predicting node labels in homophilic and heterophilic graphs, respectively. ODNet consistently ranks among the top-performing methods with minimal variance. The performance results for baseline methods are sourced from prior studies (Chamberlain et al., 2021; Chien et al., 2021; Wang et al., 2022). Notably, our model outperforms other continuous message pass-

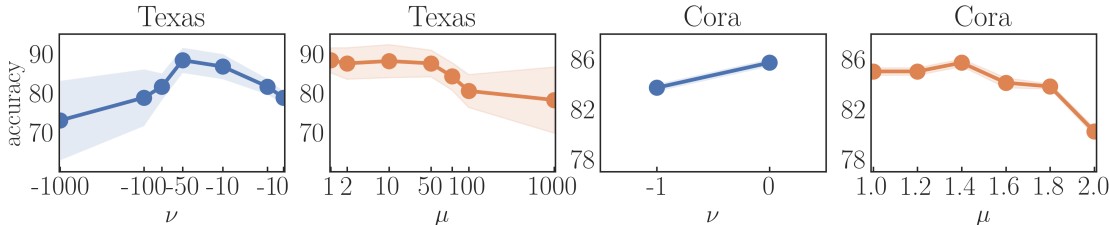

Figure 2: Impact of scaling factors $\nu$ and $\mu$ on **Texas** and **Cora**.

Table 2: Average test accuracy on **heterophilic** graphs over 10 fixed split datasets.

|  | **Texas** | **Wisconsin** | **Cornell** |
|---|---|---|---|
| homophily level | 0.11 | 0.21 | 0.30 |
| MLP | 80.8±4.8 | 85.3±3.3 | 81.9±6.4 |
| GCN | 55.1±5.2 | 51.8±3.1 | 60.5±5.3 |
| GAT | 52.2±6.6 | 49.4±4.1 | 61.9±5.1 |
| GPRGNN | 78.4±4.4 | 82.9±4.2 | 80.3±8.1 |
| H2GCN | 84.9±7.2 | 87.7± 5.0 | 82.7±5.3 |
| GCNII | 77.6±3.8 | 80.4±3.4 | 77.9±3.8 |
| Geom-GCN | 66.8±2.7 | 64.5±3.7 | 60.5±3.7 |
| PairNorm | 60.3±4.3 | 48.4±6.1 | 58.9±3.2 |
| GraphSAGE | 82.4±6.1 | 81.2±5.6 | 76.0±5.0 |
| GraphCON | 85.4±4.2 | 87.8±3.3 | 84.3±4.8 |
| ODNet | **88.3±3.2** | **89.1± 2.9** | **86.5±5.5** |

ing techniques, such as GRAND, by introducing the potential term and the bounded confidence mechanism. This superiority is particularly evident on heterophilic graphs, where the repulsive force among dissimilar node pairs significantly enhances prediction accuracy. Furthermore, Figure 2 illustrates the distinct preferences of the influence function for homophilic and heterophilic graphs. Based on the theoretical analysis and empirical evidence, we recommend following (8) for the former and (10) for the latter in general. The similarity cutoff also exhibits differing preferences. In homophilic graphs, nodes tend to amplify attraction among similar entities, while in heterophilic graphs, dissimilar nodes benefit more from emphasizing repulsion. Additional evidence is provided in Section 6.5. Furthermore, the introduced bounded confidence in ODNet, similar to an activation function, does not increase the computational cost. For instance, training one round of **CORA** (on a single A100 card) requires approximately 120 seconds for ODNet, 135 seconds for GRAND, and 164 seconds for SINN.

### 6.3 Node Classification on Hypergrpahs

**Benchmark Datasets** The hypergraph variant of ODNet undergoes an evaluation through semi-supervised node classification tasks conducted on four benchmark hypergraphs extracted from Yadati et al. (2019). For co-citation networks (**Cora-cocitation**, **CiteSeer-cocitation**, and **PubMed-cocitation**), documents cited by a given document are interconnected by a hyperedge. Similarly, the co-authorship networks (**Cora-coauthor**) aggregates all documents co-authored by an individual into a single hyperedge. A statistical summary of the four benchmarks is provided in Table 5 in Appendix B.

**Result Analysis** In contrast to graph data with relatively sparse connections, hypergraphs utilize a few hyperedges and establish densely connected local communities. As reported in Table 3, ODNet consistently outperforms baseline methods with a significant improvement. It is worth noting that our ODNet adopts the hypergraph weights $a_{ij}^e$ from HGNN (Feng et al., 2019) with a simple Euler scheme of first-order forward differences. In this sense, our method demonstrates great potential for significantly enhancing the performance of a basic method with minimal additional complexity, surpassing even more advanced methods.

Table 3: Average test accuracy on **hypergraphs** over 10 random splits.

|  | **Cora**-coauthor | **Cora**-cocitation | **CiteSeer**-cocitation | **PubMed**-cocitation |
|---|---|---|---|---|
| HGNN | 82.6±1.7 | 79.4±1.4 | 72.5±1.2 | 86.4±0.4 |
| HYPERGCN | 79.5±2.1 | 78.5±1.3 | 71.3±0.8 | 82.8±8.7 |
| HCHA | 82.6±1.0 | 79.1±1.0 | 72.4±1.4 | 86.4±0.4 |
| HNHN | 77.2±1.5 | 76.4±1.9 | 72.6±1.6 | 86.9±0.3 |
| UNIGCNII | 83.6±1.1 | 78.8±1.1 | 73.0±2.2 | 88.3±0.4 |
| HYPERND | 80.6±1.3 | 79.2±1.1 | 72.6±1.5 | 86.7±0.4 |
| ALLDEEPSETS | 82.0±1.5 | 76.9±1.8 | 70.8±1.6 | 88.8±0.3 |
| ALLSETTRANS | 83.6±1.5 | 78.6±1.5 | 73.1±1.2 | 88.7±0.4 |
| SheafHGCN | 83.2±1.2 | 80.1±1.1 | 73.2±0.5 | 87.1±0.7 |
| ED-HNN | 84.0±1.6 | 80.3±1.4 | 73.7±1.4 | 89.0±0.5 |
| ODNET (Ours) | **84.5±1.6** | **80.7±0.9** | **74.0±0.9** | **89.0±0.4** |

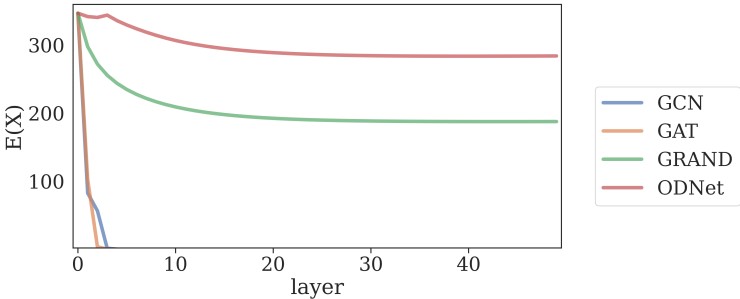

Figure 3: Decays of Dirichlet energy with layers on **Texas**.

## 6.4 Dirichlet Energy and Community Consensus

Many message passing methods encounter the issue of oversmoothing, limiting their ability to enable deep networks to achieve expressive propagation. A GNN model is considered to alleviate the oversmoothing problem if its Dirichlet energy rapidly approaches a lower bound as the number of network layers increases (Cai & Wang, 2020). Figure 3 illustrates the decay of Dirichlet energy on **Texas** with all network parameters randomly initialized. The two conventional message passings, GCN and GAT, exhibit a sudden progression of Dirichlet energy with exponential decay. In contrast, GRAND employs a small multiplier to delay all nodes' features to collapse to the same value. ODNET stabilizes the energy decay with bounded confidence and the influence weights, offering a simple and efficient solution to alleviate the oversmoothing issue. Since stable Dirichlet energy reflects the disparity of feature clusters, the observation that a decreasing profile of $\phi$ reduces Dirichlet energy under stable conditions is consistent with simulation results in opinion dynamics that heterophily dynamics enhances consensus (Motsch & Tadmor, 2014). Note that the 'heterophily' in opinion dynamics refers to the tendency of a graph to form stronger connections with those who are different rather than those who are similar, which is a different term from the 'heterophilic graph' in GNNs.

## 6.5 Additional Investigation

**Neural ODE Solvers** The Dormand–Prince adaptive step size scheme (DOPRI5) served as the neural ODE solver for ODNET. Additionally, we evaluated the performance of two other solvers across various datasets: the Runge-Kutta method (rk4) and the first-order Euler scheme (Euler). The outcomes are presented in Figure 4. Although different solvers did not consistently demonstrate a significant and sustained advantage of one over the others, our selection of DOPRI5 yielded the overall best results.

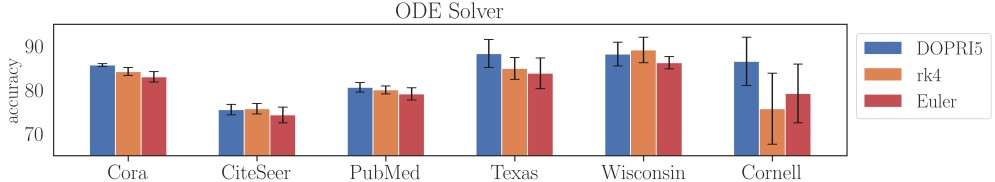

Figure 4: Prediction performance with different neural ODE solvers.

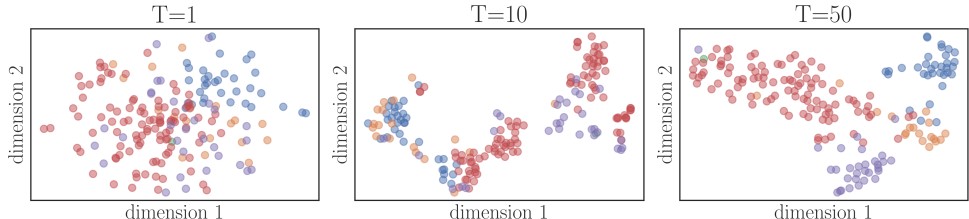

Figure 5: t-SNE visualization on node embeddings for **Texas**.

**Embedding Dynamics**   We employ the t-SNE algorithm to visualize the embedded features in a two-dimensional plane for the **Texas** dataset. We choose the output embeddings from the last layer at different epochs. Figure 5 demonstrates an evident clustering trend as the number of training epochs increases. By the 50th epoch, nodes with different labels are distinctly separable in the reduced two-dimensional space.

**Influence Function**   We next delve into the impact of different selections of the influence function $\phi$ on the performance of ODNet. This investigation encompasses various aspects, including the choice of scaling factors ($\mu$ and $\nu$) and the definition of the similarity cutoffs ($\epsilon_1$ and $\epsilon_2$). Our findings are meticulously detailed in Tables 8-9 in Appendix B, with a particular focus on the homophilic graph (**Cora**) and the heterophilic graph (**Texas**), respectively. An interesting trend emerges concerning the parameter $\nu$, indicating a clear preference. Specifically, it is advisable to incorporate a repulsive effect on heterophilic graphs by assigning a negative value to $\nu$. Conversely, for homophilic graphs, where similarity plays a pivotal role, setting $\nu = 0$ is more appropriate. To provide a direct comparison, Figure 6 showcases ODNet's performance under different similarity cutoffs, *i.e.*, $\epsilon_1$ and $\epsilon_2$. For **Cora**, we maintain $\mu = 1.4$ and $\nu = 0$, while for **Texas**, we set $\mu = 1.0$ and $\nu = -50$. Generally, in the context of homophilic graphs, setting a relatively small value for $\epsilon_2$ tends to expand the region of nodes considered similar. This approach may be beneficial in mitigating the oversmoothing issue. Conversely, for heterophilic graphs, it is advisable to reverse information from only highly similar nodes, reflected in the choice of a larger $\epsilon_2$ value (up to 0.8). However, it is crucial to exercise caution when pushing $\epsilon_2$ towards 1.0, as a discernible reduction in performance becomes evident.

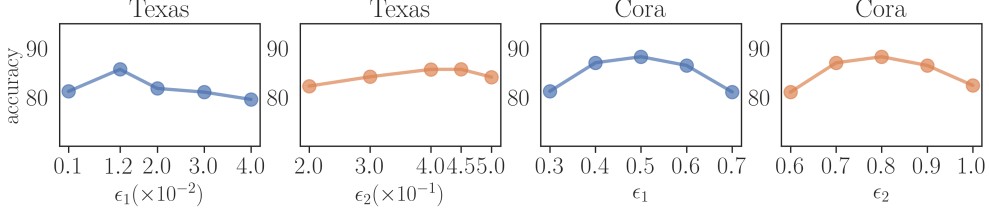

Figure 6: The impact of different $\epsilon_1$ and $\epsilon_2$ on ODNet.

## 6.6   Case Study: Biological Social Network Architecture Simplification

Microorganisms are the most extensively distributed and numerous group on Earth, which thrive in a wide array of moderate and extreme environments, such as deep-sea hydrothermal vents, ocean trenches, and

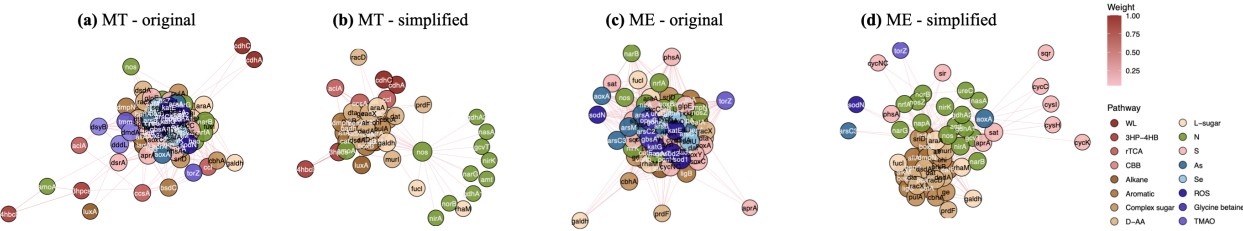

Figure 7: Co-occurrence network of selected metabolic genes in ME or MT before (a-b) and after (c-d) being simplified by ODNET. Connections are considered strong with weights> 0.05, and genes without a strong connection with any other peers are removed.

plateaus (Shu & Huang, 2022). They present the greatest diversity of life on Earth. The remarkable diversity among microorganisms finds its primary expression through their intricate metabolic pathways (Louca et al., 2018; Coelho et al., 2022). Consequently, investigating the connections between microbial metabolism in distinct environments carries profound significance in unraveling the intricate interplay between Earth's diverse ecosystems and the lives inhabiting them. Metagenomic analyses have revealed the remarkable complexity inherent to metabolic gene networks, due to the diversity and richness of functional genes and their interconnections. It thus becomes a necessity to simplify metabolic gene networks for investigating relationships among functional genes and key genes. Presently, the prevailing approach involves adjusting connection weights to streamline the network, often relying on biological expertise. However, the absence of a standardized simplification criterion results in a heavy bias in network structures influenced by the subjective opinions of biologists.

**Problem Formulation and Training Setup** As an example of the environmental microbiome analysis, the co-occurrence network is challenging to interpret due to the massive and complicated characteristics of genes and the unclear standard for assessment. The target here is to learn meaningful influence weights between gene pairs that simplify the co-occurrence network with effective biological justification. To this end, two networks originated from the microbial comparison between the *Mariana Trench* (MT) and *Mount Everest* (ME) networks (Liu et al., 2022) are utilized, where nodes are functional genes and edges are weighted by the probability of two key functional gene sets simultaneously occurring in the same species. Edges with exceptionally small weights will be discarded as noisy observations. As we are eager to identify the key genes and gene clusters from gene interactions, we construct the graph with initial connectivity (edges and edge weights), leaving any node attributes (*e.g.*, function and pathway) unobserved. We define a node-level classification task for predicting whether a node is a 'strong', 'medium', or 'weak' influencer to its community, where the three levels are cut by their degree. Further details are provided in Appendix D.

**Result Analysis** We trained two independent ODNETs on ME and MT networks, which achieved prediction accuracy as high as 96.9% and 75.0%, respectively. For both networks, metabolic genes were classified based solely on topological information, without the introduction of any a priori node features. Figure 7 visualizes the two networks in their original and the simplified appearance, respectively. For all the networks, an edge weight cutoff of 0.05 was applied to eliminate weak connections that could not be distinguishable from background noise. The original network without any simplification appeared cluttered and difficult to interpret (Figure 7a-b). In contrast, the simplified networks greatly enhanced the readability of the co-occurrence network while retaining reasonable biological significance (Figure 7c-d). Furthermore, the simplified network was able to identify the biologically key genes that acted as "opinion leaders", serving as bridges connecting different metabolic pathways. For example, in the MT network, the key gene nitrous oxide reductase (nos) bridged the carbon (Alkane, Aromatic, Complex sugar, D-AA and L-sugar) and nitrogen metabolism (N) in Figure 7c, whereas in the ME network, the key genes of sulfate reduction (sat and aprA) coupled the carbon and sulfur metabolism (S) (Figure 7d). Thus, ODNET could be employed to present more discernible networks in environmental microbiome studies, and aid in comprehending key metabolic functions within microbiomes from diverse environments.

# 7 Conclusion

This study establishes intriguing connections between sociodynamics and graph neural networks, two distinct fields that both actively investigate social networks from different perspectives. By bridging concepts from these two fields, we introduce bounded confidence for neural message passing, a novel mechanism inspired by opinion dynamics. The proposed ODNET effectively addresses the oversmoothing issue and consistently achieves top-notch performance in node prediction tasks across graphs with diverse local connectivity patterns. This success is attributed to the simplicity and efficacy of our piecewise message propagation rule.

In addition to providing strong theoretical support and empirical performance, ODNET integrates theories from social networks, offering intuitive explanations for hyperparameter settings and the significance of network parameter learning. This integration showcases ODNET's significant potential in simplifying complex real-world social networks, presenting a fresh analytical approach that does not rely on existing attributive classification conventions. The robust performance of ODNET extends its applicability to simplifying intricate networks containing a wealth of biological information, such as genes, gene-gene interactions, and metabolic pathways. This method's exceptional capacity to extract accurate insights and unveil the intrinsic mechanisms of cellular physiology provides invaluable support to biologists in deciphering the mechanisms of adaptation and pathway functions in the microbial realm. These findings are significant in understanding the interactions between Earth's environments and the metabolism of life.

### Acknowledgments

The authors acknowledge support from the National Natural Science Foundation of China (62302291, 92451301).

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
