## A   Theoretical results and proof

In this section, we prove that for our proposed ODNET on hypergraph convolution, the Dirichlet energy does not decay to zero. In other words, the proposed propagation rule mitigates the oversmoothing issue.

We set $\{u_i\} := \{x_i | i \in \mathcal{I}_1\}$ and $\{v_j\} := \{x_j | j \in \mathcal{I}_2\}$. Then we can rewrite (15) as

$$
\begin{cases}
\dot{u}_i = \dfrac{1}{N_1} \sum_{i'=1}^{N_1} a_{i,j}^{e,+}(u_{i'} - u_i) - \dfrac{1}{N_2} \sum_{j=1}^{N_2} a_{i,j}^{e,-}(v_j - u_i) + \delta u_i(1 - (u_i)^2) \\
\dot{v}_j = \dfrac{1}{N_2} \sum_{j'=1}^{N_2} a_{i,j}^{e,+}(v_{j'} - v_i) - \dfrac{1}{N_1} \sum_{i=1}^{N_1} a_{i,j}^{e,-}(u_i - v_j) + \delta v_j^{(}1 - (v_j)^2)
\end{cases}
\tag{16}
$$

In the following proof, we will use $x$ to replace $\mathbf{x}$ by $x$ to consider channel-wise variables. For clarity, we define the following notations:

- $M_2(U) := \sum\limits_{i=1}^{N_1} u_i^2, \quad M_2(V) := \sum\limits_{j=1}^{N_2} v_j^2;$

- $\widehat{M_2} := M_2(\hat{U}) + M_2(\hat{V}) = \mathrm{var}(u) + \mathrm{var}(v);$

- $\psi_i^{e,\pm} := \sum\limits_{j \in e} a_{i,j}^{e,\pm}, \; \psi_i^{\pm} := \sum\limits_{e \in \mathcal{E}} \sum\limits_{j \in e} a_{i,j}^{e,\pm};$

- $k := \max\limits_{k} \{|\mathcal{E}(k)|\};$

- $D_e^- := \max\limits_{k} \{\psi_k^{e,-}\}, \; D^- := \max\limits_{e} \{D_e^-\};$

- $D_2^- := \max\limits_{e} \{\|\psi^{e,-}\|_2\}.$

We assume $N_1 = N_2 := N_0$, which implies $N_1$ is comparable to $N_2$, *i.e.*, there exists a positive constant $\kappa$ satisfying $\frac{1}{\kappa} N_1 \leq N_2 \leq \kappa N_1$.

**Lemma 2** ($L_2$ estimate for $M_2$). *There exists a positive constant $M_2^\infty$ such that*

$$
\sup_{0 \leq t < \infty} \leq M_2^\infty \leq \infty.
\tag{17}
$$

*Proof.* We start from the derivatives of $dt M_2(U)$ and $dt M_2(V)$. For the former,

$$
\begin{aligned}
\frac{d}{dt} M_2(U) &= \frac{2}{N_1} \sum_{i=1}^{N_1} u_i \dot{u}_i \\
&= \frac{2}{N_1} \sum_{i=1}^{N_1} \sum_{e \in \mathcal{E}(i)} \sum_{i' \in e} a_{i,i'}^{e,+}(u_{i'} - u_i)^2 - \frac{2}{N_1} \sum_{i=1}^{N_1} \sum_{e \in \mathcal{E}(i)} \sum_{j \in e} a_{i,j}^{e,-}(v_j - u_i)u_i + \frac{2\delta}{N_1} \sum_{i=1}^{N_1} (u_i)^2(1 - (u_i)^2) \\
&= \frac{2}{N_1} \sum_{e \in \mathcal{E}} \sum_{i,i' \in e} a_{i,i'}^{e,+}(u_{i'} - u_i)u_i - \frac{2}{N_1} \sum_{e \in \mathcal{E}} \sum_{i,j \in e} a_{i,j}^{e,-}(v_j - u_i)u_i + \frac{2\delta}{N_1} \sum_{i=1}^{N_1} (u_i)^2(1 - (u_i)^2) \\
&= -\frac{1}{N_1} \sum_{e \in \mathcal{E}} \sum_{i,,i' \in e}^{N_1} a_{i,i'}^{e,+}(u_{i'} - u_i)^2 - \frac{2}{N_1} \sum_{e \in \mathcal{E}} \sum_{i,j \in e} a_{i,j}^{e,-}(v_j - u_i)u_i + \frac{2\delta}{N_1} \sum_{i=1}^{N_1} (u_i)^2(1 - (u_i)^2).
\end{aligned}
\tag{18}
$$

Similarly,

$$
\begin{aligned}
\frac{d}{dt}M_2(W) &= \frac{2}{N_2}\sum_{j=1}^{N_2} v_j \dot{v}_j \\
&= \frac{1}{N_2}\sum_{e\in\mathcal{E}}\sum_{j,,j'\in e} a_{i,i'}^{e,+}(v_{j'}-v_j)^2 - \frac{2}{N_2}\sum_{e\in\mathcal{E}}\sum_{j,i\in e} a_{j,i}^{e,-}(u_i-v_j)v_j + \frac{2\delta}{N_3}\sum_{j=1}^{N_2}(v_j)^2(1-(v_j)^2).
\end{aligned}
\tag{19}
$$

Note that $a_{i,j}^{e,\pm} = a_{j,i}^{e,\pm}$. Sum the $M_2(U)$ and $M_2(V)$, then

$$
\begin{aligned}
\frac{d}{dt}M_2 \leq &-\frac{2}{N_1}\sum_{e\in\mathcal{E}}\sum_{i,j\in e} a_{i,j}^{e,-}(v_j-u_i)u_i - \frac{2}{N_2}\sum_{e\in\mathcal{E}}\sum_{j,i\in e} a_{j,i}^{e,-}(u_i-v_j)v_j \\
&+\frac{2\delta}{N_1}\sum_{i=1}^{N_1}(u_i)^2(1-(u_i)^2) + \frac{2\delta}{N_2}\sum_{i=1}^{N_2}(v_i)^2(1-(v_i)^2).
\end{aligned}
\tag{20}
$$

By the Cauchy-Schwarz inequality,

$$
\left(\sum_{i=1}^{N_1}(u_i)^2\right)^2 \leq N_1\sum_{i=1}^{N_1}(u_i)^4, \quad \left(\sum_{i=1}^{N_1}(u_i)^2\right)^2 \leq N_2\sum_{i=1}^{N_2}(v_i)^4,
$$
$$
(u_i-v_j)^2 \leq 2((u_i)^2+(v_j)^2).
$$

Then we have

$$
\begin{aligned}
\frac{d}{dt}M_2 \leq &\frac{D^-}{N_1}\sum_{e\in\mathcal{E}}\sum_{i,j\in e}\left((v_j-u_i)^2+u_i^2\right) + \frac{D^-}{N_2}\sum_{e\in\mathcal{E}}\sum_{j,i\in e}\left((u_i-v_j)^2+v_j^2\right) \\
&+\frac{2\delta}{N_1}\sum_{i=1}^{N_1}(u_i)^2(1-(u_i)^2) + \frac{2\delta}{N_2}\sum_{i=1}^{N_2}(v_i)^2(1-(v_i)^2) \\
\leq &\frac{D^-}{N_1}\sum_{e\in\mathcal{E}}\sum_{i,j\in e}\left(3u_i^2+2v_j^2\right) + \frac{D^-}{N_2}\sum_{e\in\mathcal{E}}\sum_{i,j\in e}\left(2u_i^2+3v_j^2\right) + 2\delta M_2 - \delta M_2^2.
\end{aligned}
\tag{21}
$$

These relations yield a Riccati-type differential inequality:

$$
\begin{aligned}
\frac{d}{dt}M_2 \leq &5D^-\max\{\frac{1}{N_1},\frac{1}{N_2}\}\sum_{e\in\mathcal{E}}\sum_{i,j\in e}\left(u_i^2+v_j^2\right) + 2\delta M_2 - \delta(M_2)^2 \\
\leq &5D^-k\max\{\frac{1}{N_1},\frac{1}{N_2}\}M_2 + 2\delta M_2 - \delta(M_2)^2 \\
\leq &(C_1+2\delta)M_2 - \delta(M_2)^2,
\end{aligned}
\tag{22}
$$

where

$$
C_1 := 5D^-k\max\{\frac{1}{N_1},\frac{1}{N_2}\}.
\tag{23}
$$

Let $y$ be a solution of the following ODE:

$$
y' = \alpha C_1 y - \delta y^2.
\tag{24}
$$

Then, by phase line analysis, the solution $y(t)$ to (24) satisfies

$$
M_2(t) \leq y(t) \leq \max\left\{\frac{C_1}{\delta}+2, M_2(0)\right\} =: M_2^\infty.
\tag{25}
$$

$\square$

**Lemma 3.** *Let $u, v$ be the solution to (16). Then $|\bar{u} - \bar{v}|^2$ satisfies*

$$\frac{1}{2}\frac{d}{dt}|\bar{u} - \bar{v}|^2 \geq \left(\frac{2c_m}{N_0} - c_1\right)|\bar{u} - \bar{v}|^2 - \frac{4(D_2^-)^2}{c_1 N_0}k\widehat{M_2}. \tag{26}$$

*Proof.* The time evolution of $\bar{u}$ is given by

$$
\begin{aligned}
\dot{\bar{u}} &= \frac{1}{N_1}\sum_{i=1}^{N_1}\sum_{i'\in\mathcal{E}(i)}a_{i,i'}^{e,+}(u_{i'} - u_i) - \frac{1}{N_1}\sum_{i=1}^{N_1}\sum_{j\in\mathcal{E}(i)}a_{j,i}^{e,-}(v_j - u_i) \\
&= -\frac{1}{N_1}\sum_{e\in\mathcal{E}}\sum_{i,j\in e}a_{i,j}^{e,-}(v_j - u_i) \\
&= -\frac{1}{N_1}\sum_{e\in\mathcal{E}}\psi_j^{e,-}v_j + \frac{1}{N_1}\sum_{e\in\mathcal{E}}\psi_i^{e,-}u_i \\
&= \sum_{e\in\mathcal{E}}\left(-\frac{1}{N_1}\psi_j^{e,-}v_j + \frac{1}{N_1}\psi_i^{e,-}u_i\right).
\end{aligned}
\tag{27}
$$

The first equality uses the relation $\sum_{i=1}^{N_1}\hat{u}_i = 0$. Then we have

$$
\begin{aligned}
\frac{1}{2}\frac{d}{dt}|\bar{u} - \bar{v}|^2 &= (\bar{u} - \bar{v})(\dot{\bar{u}} - \dot{\bar{v}}) \\
&= (\bar{u} - \bar{v})\left[\sum_{e\in\mathcal{E}}\left(-\frac{1}{N_1}\psi_j^{e,-}v_j + \frac{1}{N_1}\psi_i^{e,-}u_i\right) - \sum_{e\in\mathcal{E}}\left(-\frac{1}{N_2}\psi_i^{e,-}u_i + \frac{1}{N_2}\psi_j^{e,-}v_j\right)\right] \\
&= (\bar{u} - \bar{v})\sum_{e\in\mathcal{E}}\left(-\frac{1}{N_1}\psi_j^{e,-}v_j - \frac{1}{N_2}\psi_j^{e,-}v_j + \frac{1}{N_1}\psi_i^{e,-}u_i + \frac{1}{N_2}\psi_i^{e,-}u_i\right) \\
&= (\bar{u} - \bar{v})\sum_{e\in\mathcal{E}}\left(-\frac{1}{N_1}\psi_j^{e,-}(\bar{v} + \hat{v}_j) - \frac{1}{N_2}\psi_j^{e,-}(\bar{v} + \hat{v}_j) + \frac{1}{N_1}\psi_i^{e,-}(\bar{u} + \hat{u}_i) + \frac{1}{N_2}\psi_i^{e,-}(\bar{u} + \hat{u}_j)\right) \\
&= (\bar{u} - \bar{v})\left\{\left[-\left(\frac{1}{N_1} + \frac{1}{N_2}\right)\sum_{e\in\mathcal{E}}\sum_{j\in e}\psi_j^{e,-}\right]\bar{v} + \left[\left(\frac{1}{N_1} + \frac{1}{N_2}\right)\sum_{e\in\mathcal{E}}\sum_{i\in e}\psi_i^{e,-}\right]\bar{u}\right. \\
&\qquad \left.\left[-\left(\frac{1}{N_1} + \frac{1}{N_2}\right)\sum_{e\in\mathcal{E}}\sum_{j\in e}\psi_j^{e,-}\right]\hat{v}_j + \left[\left(\frac{1}{N_1} + \frac{1}{N_2}\right)\sum_{e\in\mathcal{E}}\sum_{i\in e}\psi_i^{e,-}\right]\hat{u}_i\right\} \\
&= \frac{2}{N_0}(\bar{u} - \bar{v})\left[\sum_{e\in\mathcal{E}}\sum_{i\in e}\psi_i^{e,-}\bar{u} - \sum_{e\in\mathcal{E}}\sum_{j\in e}\psi_j^{e,-}\bar{v}\right] + \frac{2}{N_0}(\bar{u} - \bar{v})\left[\sum_{e\in\mathcal{E}}\sum_{i\in e}\psi_i^{e,-}\hat{u}_i - \sum_{e\in\mathcal{E}}\sum_{j\in e}\psi_j^{e,-}\hat{v}_j\right].
\end{aligned}
\tag{28}
$$

We denote

$$\text{Pse}(\hat{u}, \hat{v}) := \sum_{e\in\mathcal{E}}\sum_{i\in e}\psi_i^{e,-}\hat{u}_i - \sum_{e\in\mathcal{E}}\sum_{j\in e}\psi_j^{e,-}\hat{v}_j$$

and

$$\text{Pse}(\bar{u}, \bar{v}) := \sum_{e\in\mathcal{E}}\sum_{i\in e}\psi_i^{e,-}\bar{u} - \sum_{e\in\mathcal{E}}\sum_{j\in e}\psi_j^{e,-}\bar{v}.$$

Assume there exist constants $c_m, c_v$, such that

$$\text{Pes}(\bar{u}, \bar{v}) \geq c_m(\bar{u} - \bar{v}). \tag{29}$$

Then, by Cauchy inequality, for any $c_1$, we have

$$
\begin{aligned}
\frac{1}{2}\frac{d}{dt}|\bar{u}-\bar{v}|^2 &= \frac{2}{N_0}\mathrm{Pes}(\bar{u},\bar{v})(\bar{u}-\bar{v}) + \frac{2}{N_0}\mathrm{Pes}(\hat{u},\hat{v})(\bar{u}-\bar{v}) \geq \left(\frac{2c_m}{N_0}-c_1\right)|\bar{u}-\bar{v}|^2 + c_1|\bar{u}-\bar{v}|^2 \\
&\quad + \frac{2}{N_0}\mathrm{Pes}(\hat{u},\hat{v})(\bar{u}-\bar{v}) \\
&\geq \left(\frac{2c_m}{N_0}-c_1\right)|\bar{u}-\bar{v}|^2 - \frac{1}{c_1}\frac{2}{N_0}\left(\mathrm{Pes}(\hat{u},\hat{v})\right)^2 \\
&\geq \left(\frac{2c_m}{N_0}-c_1\right)|\bar{u}-\bar{v}|^2 - \frac{4}{c_1 N_0}\sum_{e\in\mathcal{E}}\left[\left(\sum_{i\in e}\psi_i^{e,-}\hat{u}_i\right)^2 + \left(\sum_{j\in e}\psi_j^{e,-}\hat{v}_j\right)^2\right] \\
&\geq \left(\frac{2c_m}{N_0}-c_1\right)|\bar{u}-\bar{v}|^2 - \frac{4}{c_1 N_0}\sum_{e\in\mathcal{E}}\left[\|\psi^{e,-}\|^2\sum_{i\in e}\hat{u}_i^2 + \|\psi^{e,-}\|^2\sum_{j\in e}\hat{v}_j^2\right] \\
&\geq \left(\frac{2c_m}{N_0}-c_1\right)|\bar{u}-\bar{v}|^2 - \frac{4(D_2^-)^2}{c_1 N_0}\sum_{e\in\mathcal{E}}\left[\sum_{i\in e}\hat{u}_i^2 + \sum_{j\in e}\hat{v}_j^2\right] \\
&\geq \left(\frac{2c_m}{N_0}-c_1\right)|\bar{u}-\bar{v}|^2 - \frac{4(D_2^-)^2}{c_1 N_0}k[\widehat{M_2}]
\end{aligned}
\tag{30}
$$

$\square$

**Lemma 4.** *Let $u,v$ be the solution to equation 16. Then $\widehat{M_2}$ satisfies*

$$
\frac{1}{2}\frac{d}{dt}\widehat{M_2} \leq C_2\widehat{M_2} + 2c_2|\bar{u}-\bar{v}|^2,
\tag{31}
$$

*where*

$$
C_2 := -k\left(C^A - 2D^- + \frac{(D^-)^2}{4c_2} - \frac{\delta}{k}\right),
\tag{32}
$$

*and $c_2$ is an arbitrary positive constant.*

*Proof.* Subtracting equation 27 gives $\dot{\hat{u}}_i$. Then we have

$$
\begin{aligned}
\frac{1}{2}\frac{d}{dt}\left(\frac{1}{N_1}\sum_{i=1}^{N_1}|\hat{u}_i|^2\right) &= \frac{1}{N_1}\sum_{i=1}^{N_1}\hat{u}_i\dot{\hat{u}}_i \\
&= \frac{1}{N_1}\sum_{i=1}^{N_1}\hat{u}_i\sum_{e\in\mathcal{E}(i)}\left[\sum_{i'\in e}a_{i,i'}^{e,+}(u_{i'}-u_i) - \sum_{j\in e}a_{i,j}^{e,-}(v_j-u_i)\right] + \frac{\delta}{N_1}\sum_{i=1}^{N_1}\hat{u}_i u_i(1-u_i^2) \\
&= \frac{1}{N_1}\sum_{i=1}^{N_1}\hat{u}_i\sum_{e\in\mathcal{E}(i)}\left[\sum_{i'\in e}a_{i,i'}^{e,+}(\hat{u}_{i'}-\hat{u}_i) - \sum_{j\in e}a_{i,j}^{e,-}(v_j-u_i)\right] + \frac{\delta}{N_1}\sum_{i=1}^{N_1}\hat{u}_i u_i(1-u_i^2) \\
&= \frac{1}{N_1}\sum_{i=1}^{N_1}\hat{u}_i\sum_{e\in\mathcal{E}(i)}\sum_{i'\in e}a_{i,i'}^{e,+}(\hat{u}_{i'}-\hat{u}_i) - \frac{1}{N_1}\sum_{i=1}^{N_1}\hat{u}_i\sum_{e\in\mathcal{E}(i)}\sum_{j\in e}a_{i,j}^{e,-}(\hat{v}_j-\hat{u}_i) \\
&\quad - \frac{1}{N_1}\sum_{i=1}^{N_1}\hat{u}_i\sum_{e\in\mathcal{E}(i)}\sum_{j\in e}a_{i,j}^{e,-}(\bar{v}-\bar{u}) + \frac{\delta}{N_1}\sum_{i=1}^{N_1}\hat{u}_i u_i(1-u_i^2) \\
&=: I_1 + I_2 + I_3 + I_4.
\end{aligned}
\tag{33}
$$

In particular,

$$
\begin{aligned}
I_1 &= \frac{1}{N_1} \sum_{i=1}^{N_1} \hat{u}_i \sum_{e \in \mathcal{E}(i)} \sum_{i' \in e} a_{i,i'}^{e,+} (\hat{u}_{i'} - \hat{u}_i) \\
&= \frac{1}{N_1} \sum_{e \in \mathcal{E}} \sum_{i,i' \in e} a_{i,i'}^{e,+} (\hat{u}_{i'} - \hat{u}_i) \hat{u}_i \\
&= \frac{1}{N_1} \sum_{e \in \mathcal{E}} (\hat{u}^e)^\top A^e \hat{u}^e,
\end{aligned}
\tag{34}
$$

where $\hat{u}^e := (\hat{u}_{i_1}, \cdots, \hat{u}_{i_{|e|}})^\top$ for each $e$. Thus $I_1$ is bounded by

$$
\frac{1}{N_1} \sum_{e \in \mathcal{E}} (\hat{u}^e)^\top A^e \hat{u}^e \leq -\frac{1}{N_1} \sum_{e \in \mathcal{E}} C^A |\hat{u}^e|^2 \leq -\frac{c_r}{N_1} \sum_{i=1}^{N_1} C^A |\hat{u}_i|^2,
\tag{35}
$$

where $c_r$ is a constant larger than 1 related to the repetition of $\{\hat{u}_i\}$ in all hyperedges.

$I_2$ can be controlled by

$$
\begin{aligned}
I_2 &= -\frac{1}{N_1} \sum_{i=1}^{N_1} \hat{u}_i \sum_{e \in \mathcal{E}(i)} \sum_{j \in e} a_{i,j}^{e,-} (\hat{v}_j - \hat{u}_i) \\
&= -\frac{1}{N_1} \sum_{e \in \mathcal{E}} \sum_{i,j \in e} a_{i,j}^{e,-} (\hat{v}_j - \hat{u}_i) \hat{u}_i \\
&= -\frac{1}{N_1} \sum_{e \in \mathcal{E}} \sum_{i,j \in e} a_{i,j}^{e,-} \hat{v}_j \hat{u}_i + \frac{1}{N_1} \sum_{e \in \mathcal{E}} \sum_{i,j \in e} a_{i,j}^{e,-} |\hat{u}_i|^2 \\
&\leq \frac{1}{N_1} \sum_{e \in \mathcal{E}} \sum_{i,j \in e} a_{i,j}^{e,-} \frac{1}{2} (|\hat{v}_j|^2 + |\hat{u}_i|^2) + \frac{1}{N_1} \sum_{e \in \mathcal{E}} D_e^- \sum_{i \in e} |\hat{u}_i|^2 \\
&\leq \frac{1}{2N_1} \sum_{e \in \mathcal{E}} \sum_{j \in e} \psi_j^{e,-} |\hat{v}_j|^2 + \frac{3D^-}{2N_1} \sum_{e \in \mathcal{E}} \sum_{i \in e} |\hat{u}_j|^2 \\
&\leq \frac{D^-}{2N_1} \sum_{e \in \mathcal{E}} \sum_{j \in e} |\hat{v}_j|^2 + \frac{3D^-}{2N_1} \sum_{e \in \mathcal{E}} \sum_{i \in e} |\hat{u}_j|^2 \\
&\leq \frac{D^- k}{2N_1} \sum_{j=1}^{N_2} |\hat{v}_j|^2 + \frac{3D^- k}{2N_1} \sum_{i=1}^{N_1} |\hat{u}_i|^2.
\end{aligned}
\tag{36}
$$

$I_3$ has the below estimate for any constant $c_2 > 0$:

$$
\begin{aligned}
I_3 &= -\frac{1}{N_1} \sum_{i=1}^{N_1} \hat{u}_i \sum_{e \in \mathcal{E}(i)} \sum_{j \in e} a_{i,j}^{e,-} (\bar{v} - \bar{u}) \\
&\leq c_2 |\bar{u} - \bar{v}|^2 + \frac{1}{4 c_2 N_1} \sum_{e \in \mathcal{E}} \sum_{i,j \in e} |a_{i,j}^{e,-}|^2 |\hat{u}_i|^2 \\
&\leq c_2 |\bar{u} - \bar{v}|^2 + \frac{(D^-)^2}{4 c_2 N_1} \sum_{e \in \mathcal{E}} \sum_{i,j \in e} |\hat{u}_i|^2 \\
&\leq c_2 |\bar{u} - \bar{v}|^2 + \frac{(D^-)^2 k}{4 c_2 N_1} \sum_{e \in \mathcal{E}} \sum_{i=1}^{N_1} |\hat{u}_i|^2.
\end{aligned}
\tag{37}
$$

Define

$$
I_4 := \frac{1}{N_1} \delta \sum_{i=1}^{N_1} \hat{u}_i u_i (1 - |u_i|^2).
\tag{38}
$$

$$I_4 = \frac{\delta}{N_1} \sum_{i=1}^{N_1} \hat{u}_i(\hat{u}_i + \bar{u})(1 - u_i^2)$$

$$= \frac{\delta}{N_1} \sum_{i=1}^{N_1} \hat{u}_i^2 - \frac{\delta}{N_1} \sum_{i=1}^{N_1} \hat{u}_i^2 u_i^2 + \frac{\delta}{N_1} \sum_{i=1}^{N_1} \hat{u}_i \bar{u} - \frac{\delta}{N_1} \sum_{i=1}^{N_1} \hat{u}_i u_i^2 \bar{u} \tag{39}$$

$$= \frac{\delta}{N_1} \sum_{i=1}^{N_1} \hat{u}_i^2 - \frac{\delta}{N_1} \sum_{i=1}^{N_1} \hat{u}_i |u_i|^2 u_i$$

Note that

$$\sum_{i=1}^{N_1} \hat{u}_i |u_i|^2 u_i = \sum_{i=1}^{N_1} |u_i|^2 (|u_i|^2 - u_i \bar{u})$$

$$\geq \frac{1}{2} \sum_{i=1}^{N_1} |u_i|^2 (|u_i|^2 - |\bar{u}|^2)$$

$$= \frac{1}{2} \sum_{i=1}^{N_1} |u_i|^4 - \frac{1}{2} \sum_{i=1}^{N_1} |u_i|^2 |\bar{u}|^2 \tag{40}$$

$$\geq \frac{1}{2} \sum_{i=1}^{N_1} |u_i|^4 - \frac{1}{2N_1} \sum_{i=1}^{N_1} (|u_i|^2)^2$$

$$\geq 0.$$

Hence,

$$I_4 \leq \delta \widehat{M}_2(u). \tag{41}$$

Then

$$\frac{d}{dt}\left(\frac{1}{2N_1} \sum_{i=1}^{N_1} |\hat{u}_i|^2\right) \leq k\left(-\frac{C^A}{N_1} + \frac{3D^-}{2N_1} + \frac{(D^-)^2}{4c_2 N_1}\right) \sum_{i=1}^{N_1} |\hat{u}_i|^2 + k\frac{D^-}{2N_2} \sum_{j=1}^{N_2} |\hat{v}_j|^2 + c_2 |\bar{u} - \bar{v}|^2$$

$$\leq \left[k\left(-C^A + \frac{3D^-}{2} + \frac{(D^-)^2}{4c_2}\right) + \delta\right] \widehat{M}_2(u) + k\frac{D^-}{2}\widehat{M}_2(v) + c_2 |\bar{u} - \bar{v}|^2. \tag{42}$$

Similarly,

$$\frac{d}{dt}\left(\frac{1}{2N_2} \sum_{j=1}^{N_2} |\hat{v}_j|^2\right) \leq \left[k\left(-C^A + \frac{3D^-}{2} + \frac{(D^-)^2}{4c_2}\right) + \delta\right] \widehat{M}_2(v) + k\frac{D^-}{2}\widehat{M}_2(u) + c_2 |\bar{u} - \bar{v}|^2. \tag{43}$$

Sum them together. We have

$$\frac{1}{2}\frac{d}{dt}(\widehat{M}_2) \leq -k\left(C^A - 2D^- + \frac{(D^-)^2}{4c_2} - \frac{\delta}{k}\right)(\widehat{M}_2) + 2c_2 |\bar{u} - \bar{v}|^2. \tag{44}$$

This gives an exponential growth estimate or $\widehat{M}_2$ up to an error term of $|\bar{u} - \bar{v}|^2$.

Lemma 3 gives an exponential growth estimate of $|\bar{u} - \bar{v}|^2$ up to an error term of $\widehat{M}_2$. If $N_0$ is large enough, the coefficient of the error term will be small.

Lemma 4 gives an exponential growth estimate or $\widehat{M}_2$ up to an error term of $|\bar{u} - \bar{v}|^2$. If $N_0$ is large enough, the coefficient of the error term is small.

Set

$$A_{11} := \frac{2c_m}{N_0} - c_1, \; A_{12} := \frac{4(D_2^-)^2 k}{c_1 N_0}, \; A_{21} = 2c_2, \; A_{22} = k\left(C^A - 2D^- + \frac{(D^-)^2}{4c_2}\right). \tag{45}$$

If $N_0$ is large enough, (45) will be satisfied.

$$(A_{11} + A_{12})^2 - 4A_{21}A_{12} > 0. \tag{46}$$

Apply Lemma 4.1 in Jin & Shu (2021a), we can then obtain the $L_2$ separation.

The relative size between $|\bar{u} - \bar{v}|^2$ and $\widehat{M_2}$ is an indicator of group separation in the sense of $L_2$. If $|\bar{u} - \bar{v}|^2$ is much larger, $\widehat{M_2}$, then the two groups are well-separated in an average sense. Since the hypergraph is connected, there is a positive bound between different clusters, hence the Dirichlet energy does not decay to zero.

## B  Additional Experimental Details

### B.1  Baseline Methods

We compare ODNet with a range of discrete and continuous message-passing methods across homophilic, heterophilic, and hypergraphs.

**Homophilic Graphs**  We compare the performance of ODNet with classical graph convolutions such as Graph Convolutional Network (GCN; Yadati et al. (2019)), Graph Attention Network (GAT; Veličković et al. (2018)), Mixture Model Networks (MoNet; Monti et al. (2017)), and GraphSage (Hamilton et al., 2017). We also include ODE-based continuous graph convolutions, including Continuous Graph Neural Networks (CGNN; Xhonneux et al. (2020)), Graph Neural Ordinary Differential Equations (GDE; Poli et al. (2020)), and Graph Neural Diffusion with linear diffusion (GRAND-l; Chamberlain et al. (2021)). Furthermore, we include the other opinion-dynamic inspired network, SINN Okawa & Iwata (2022).

**Heterophilic Graphs**  For the heterophilic graphs, we make comparisons with representative graph neural networks that are known for encoding heterophilic graphs (GPRGNN (Chien et al., 2021), GRAPHCON (Rusch et al., 2022), and H2GCN (Zhu et al., 2020)) or for mitigating the oversmoothing issue (GCNII (Chen et al., 2020), GEOM-GCN (Pei et al., 2020), and PAIRNORM (Zhao & Akoglu, 2020)). For completeness, we also include the classic graph convolutions, including GraphSAGE (Hamilton et al., 2017), GAT (Veličković et al., 2018), and GCN (Kipf & Welling, 2017)

**Hypergraphs**  The performance of ODNET on the four hypergraphs are compared with a diverse of baselines, including HGNN (Feng et al., 2019), HYPERGCN (Yadati et al., 2019), HCHA (Bai et al., 2021), HNHN (Dong et al., 2020), UNIGCNII (Huang & Yang, 2021), HYPERND Tudisco et al. (2021), ALLDEEPSETS (Chien et al., 2022), ALLSETTRANSFORMER (Chien et al., 2022), ED-HNN (Wang et al., 2022), and SHEAFHGNN (Duta et al., 2024).

### B.2  Experimental Details

Table 4-5 reports summary statistics of the graphs and hypergraphs used in Section 6, respectively.

Table 6 provides the optimal combination of hyperparameters for each benchmark to reproduce the performance of ODNET reported in the paper.

Table 7 details the searching space of the hyperparameters. For general hyperparameters, such as learning rate and weight decay, we used Ray Tune with a hundred trials using an asynchronous hyperband scheduler with a grace period of 50 epochs. Table 8-9 reports the influence of selecting different scaling factors on the model performance. We test the impacts on **Cora** and **Texas** for both homophilic and heterophilic graphs representation learning.

## C  Additional Background: Co-Occurrence Network of Metabolic Genes

The co-occurrence network of metabolic genes is a graph representation that illustrates the statistical associations and co-occurrence patterns among various metabolic genes within a biological system (Bello et al.,

Table 4: Summary of **graph** benchmarks used in experiments.

| Dataset | # classes | # features | # nodes | # edges | $\mathcal{H}$ |
|---|---|---|---|---|---|
| **Cora** (McCallum et al., 2000) | 7 | 1,433 | 2,708 | 5,429 | 0.83 |
| **CiteSeer** (Sen et al., 2008) | 6 | 3,703 | 3,327 | 4,732 | 0.71 |
| **PubMed** (Namata et al., 2012) | 3 | 500 | 19,717 | 44,338 | 0.79 |
| **CoauthorCS** (Shchur et al., 2018) | 15 | 6,805 | 18,333 | 100,227 | 0.80 |
| **Computer** (Namata et al., 2012) | 10 | 767 | 13,381 | 245,778 | 0.77 |
| **Photo** (Namata et al., 2012) | 8 | 745 | 7,487 | 119,043 | 0.83 |
| **Texas** García-Plaza et al. (2016) | 5 | 1,703 | 183 | 309 | 0.11 |
| **Wisconsin** García-Plaza et al. (2016) | 5 | 1,703 | 251 | 499 | 0.21 |
| **Cornell** García-Plaza et al. (2016) | 5 | 1,703 | 183 | 295 | 0.30 |

Table 5: Summary of **hypergraph** benchmarks used in experiments.

| | **Cora**-coauthor | **Cora**-cocitation | **CiteSeer**-cocitation | **PubMed**-cocitation |
|---|---|---|---|---|
| # classes | 7 | 7 | 6 | 3 |
| # features | 1,433 | 1,433 | 3,703 | 500 |
| # hypernodes | 2,708 | 2,708 | 3,312 | 19,717 |
| # hyperedges | 1,072 | 1,579 | 1,079 | 7,963 |
| avg. hyperedge size | 4.2±4.1 | 3.0±1.1 | 3.2±2.0 | 4.3±5.7 |

Table 6: Optimal setting of hyperparameters in reproducing the results in Section 6.

| Dataset | $\epsilon_1$ | $\epsilon_2$ | time (T) | $\nu$ | $\mu$ |
|---|---|---|---|---|---|
| **Cora** | 0.012 | 0.40 | 12 | 0 | 1.4 |
| **CiteSeer** | 0.01 | 0.90 | 10 | 0 | 3.0 |
| **PubMed** | 0.01 | 0.40 | 20 | 0 | 2.2 |
| **CoauthorCS** | 0.01 | 0.40 | 15 | 0 | 1.7 |
| **Computer** | 0.01 | 0.50 | 15 | 0 | 5.0 |
| **Photo** | 0.01 | 0.40 | 12 | 0 | 10.0 |
| **Texas** | 0.50 | 0.80 | 12 | -50 | 1.0 |
| **Wisconsin** | 0.60 | 0.80 | 12 | -10 | 2.0 |
| **Cornell** | 0.12 | 0.40 | 12 | 0 | 2.0 |
| **Cora-coauthor** | 0 | 1 | 0.1 | 1.0 | 1.0 |
| **Cora-cocitation** | 0 | 1 | 0.1 | 1.0 | 1.0 |
| **PubMed-cocitation** | 0 | 1 | 0.1 | 1.0 | 1.0 |
| **CiteSeer-cocitation** | 0 | 1 | 0.1 | 1.5 | 1.0 |

2020). This network emerges from computational analyses of extensive genomic data, with each node denoting metabolic genes linked to distinct biochemical functions, such as sugar production and TMAO (trimethylamine N-oxide) synthesis. These connections are quantified by the likelihood of two crucial functional genes co-occurring within the same species at a given time (Liu et al., 2022). Due to the complexity of the co-occurrence network of metabolic genes in different microorganism species, which arise from a large number of genes and connections, simplifying the network enables us to identify how key metabolic genes in microorganisms can be gathered into several interdependent modules. This is significantly important for revealing the mechanism of how genes work together within metabolic pathways, how they respond to different environmental conditions, and which genes may have essential roles in specific biological processes, highlighting the

Table 7: Hyperparameter Search Space

| Hyperparameters | Search Space | Distribution |
|---|---|---|
| learning rate | $[10^{-6}, 10^{-1}]$ | log-uniform |
| weight decay | $[10^{-3}, 10^{-1}]$ | log-uniform |
| dropout rate | $[0.1, 0.8]$ | uniform |
| hidden dim | $\{64, 128, 256\}$ | categorical |

Table 8: Choices of influence function and scaling factors for **Cora**.

| $\epsilon_1$ | $\epsilon_2$ | $\mu$ | $\nu$ | **Accuracy** |
|---|---|---|---|---|
| 0.04 | 0.45 | 1.4 | 0 | 79.6±0.3 |
| 0.012 | 0.40 | 2.0 | 0 | 80.2±0.3 |
| 0.03 | 0.45 | 1.4 | 0 | 81.0±0.2 |
| 0.02 | 0.45 | 1.4 | 0 | 81.8±0.2 |
| 0.012 | 0.20 | 1.4 | 0 | 82.3±0.2 |
| 0.012 | 0.40 | 1.4 | -1 | 83.7±0.4 |
| 0.012 | 0.40 | 1.8 | 0 | 83.9±0.2 |
| 0.012 | 0.40 | 1.6 | 0 | 84.1±0.3 |
| 0.012 | 0.30 | 1.4 | 0 | 84.2±0.3 |
| 0.012 | 0.40 | 1.2 | 0 | 85.0±0.3 |
| 0.012 | 0.40 | 1.0 | 0 | 85.0±0.3 |

Table 9: Choices of influence function and scaling factors for **Texas**.

| $\epsilon_1$ | $\epsilon_2$ | $\mu$ | $\nu$ | **Accuracy** |
|---|---|---|---|---|
| 0.50 | 0.80 | 2.0 | -1000 | 73.0±10.1 |
| 0.50 | 0.80 | 1000.0 | -50 | 78.2±8.4 |
| 0.50 | 0.80 | 2.0 | -100 | 78.8±7.2 |
| 0.50 | 0.80 | 2.0 | 0 | 78.8±1.6 |
| 0.50 | 0.80 | 100.0 | -50 | 80.5±4.2 |
| 0.50 | 0.60 | 1.0 | -50 | 81.0±3.0 |
| 0.70 | 0.80 | 1.0 | -50 | 81.1±4.2 |
| 0.50 | 0.80 | 2.0 | -80 | 81.6±3.4 |
| 0.50 | 0.80 | 2.0 | -1 | 81.6±2.0 |
| 0.60 | 0.80 | 1.0 | -50 | 86.5±3.5 |
| 0.50 | 0.90 | 1.0 | -50 | 86.5±3.0 |
| 0.50 | 0.70 | 1.0 | -50 | 87.0±3.4 |
| 0.50 | 0.80 | 2.0 | -10 | 86.7±3.2 |
| 0.40 | 0.80 | 1.0 | -50 | 87.0±3.0 |
| 0.50 | 0.80 | 2.0 | -50 | 87.6±4.0 |
| 0.50 | 0.80 | 2.0 | -50 | 87.6±4.0 |
| 0.50 | 0.80 | 10.0 | -50 | 88.1±4.2 |
| 0.50 | 0.80 | 1.0 | -50 | 88.3±3.2 |

significance of the adaptation of microorganisms to changing environmental conditions (Levy & Borenstein, 2013).

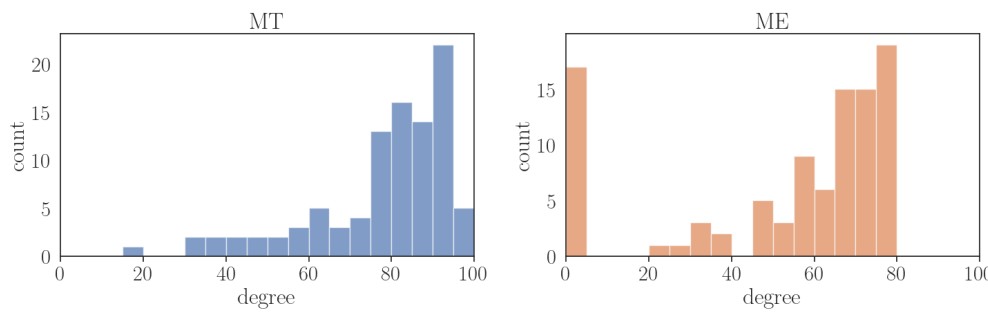

Figure 8: Distribution of node degree on **ME** and **MT** networks.

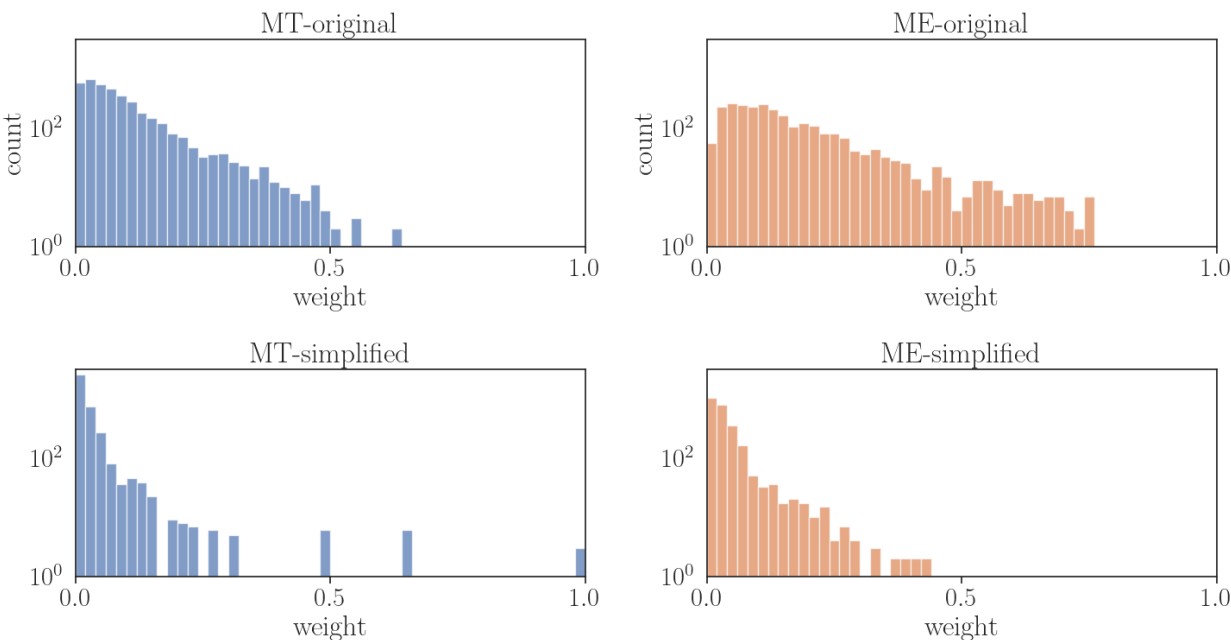

Figure 9: Distribution of edge weights on **ME** and **MT** networks.

# D   Experimental Details for the Co-Occurrence Network Simplification Task

We established two distinct graphs for the Mariana Trench (MT) and Mount Everest (ME) gene co-occurrence networks, utilizing source data from Liu et al. (2022). In both networks, the nodes represent the same set of functional genes. The primary difference between them lies in the edge weights, which reflect unique co-occurrence patterns of gene pairs in MT and ME. This distinction is visually evident when comparing the top two histograms in Figure 9, illustrating the varying distributions of gene influence weights in MT and ME graphs.

To construct these graphs, we connected all node pairs with non-zero edge weights, resulting in a total of $2,517$ edges for 96 nodes. Each node was associated with a 20-dimensional unit vector as pseudo-features. Additionally, we assigned a three-class categorical label to each node, categorizing them as 'strong,' 'medium,' or 'weak' influencers within their respective local communities. The label assignment was determined based on the nodes' degrees with cutoffs at 20 and 60. For example, a node with a degree of 30 would be classified as a 'medium influencer.'

To facilitate model training, we applied random masking to the training, validation, and test sets, ensuring equal proportions in each set. During the training process, we recorded the learned similarity scores $s_{ij}$ at the final layer for later use in generating the simplified network.

Figure 9 highlights a noticeable divergence between the top two histograms (representing the weight distributions in the original networks) and the bottom two histograms (depicting the weight distributions in the simplified networks). Specifically, a higher concentration of weights is observed at the extreme regions (with weights close to 0 and 1) in the simplified networks. This divergence underscores the impact of our simplification approach on the network's edge weight distribution.