# OpenReview forum: "ODNet: Opinion Dynamics-Inspired Neural Message Passing for Graphs and Hypergraphs"
_TMLR — Accepted by TMLR_

### Review · Reviewer_YSQv · 2024-08-29

**Summary Of Contributions:**

This paper adopts concepts from opinion dynamics models, such as the French-DeGroot Model and the Hegselmann-Krause Model, to propose a new message-passing method for graph neural networks (GNNs). The main objective is to address the oversmoothing issue in GNNs. Experimental results demonstrate that the proposed ODNet outperforms baseline methods on several datasets specific to node classification.

**Audience:**

Yes

**Broader Impact Concerns:**

None.

**Claims And Evidence:**

Yes

**Requested Changes:**

Please see above.

**Strengths And Weaknesses:**

Strengths
- The connection between opinion dynamics and message-passing is interesting.
- Promising improvements.

Weaknesses
- I recommend adding an additional paragraph in the experiment section to introduce all the baselines. In the current version, it’s challenging to link the baselines to their references and to clearly understand the key differences between the proposed method and the baselines.
- Most of the baselines are relatively outdated. It would be beneficial to consider more advanced methods, such as ED-HNN [1]. I recommend conducting a more comprehensive literature review and including newer approaches for comparison.

[1] Equivariant Hypergraph Diffusion Neural Operators, ICLR 2023.

---

> ### Author Response · Authors · 2024-11-14
>
> Thank you for your positive feedback and for recognizing the contributions and novelty of our work. We have followed your suggestions and revised the manuscript. The updates are highlighted in red in the revision. In summary:
>
> 1. **Additional Introduction to the Baseline Methods**: Thank you for your suggestion. Due to the 12-page limit in the initial submission, we could not add more detailed descriptions of the baseline methods in the main text. Instead, we included relevant explanations and discussions in the updated Appendix B.1.
>
> 2. **More Baseline Methods**: In the revision, we incorporate suggestions from the reviews and expand the baseline methods by including SheafHGCN and ED-HNN in the hypergraph section (Table 4).

---

### Review · Reviewer_k7Qs · 2024-09-21

**Summary Of Contributions:**

The paper explores connections between opinion dynamics in social networks and neural message passing in GNNs. By leveraging insights from social networks, the paper introduces a new mechanism for neural message passing that has a strong theoretical foundation. It effectively addresses the well-known problem of oversmoothing and achieves top performance in node prediction tasks across a variety of connectivity patterns.

**Audience:**

Yes

**Broader Impact Concerns:**

No concerns on the ethical implications of the work.

**Claims And Evidence:**

Yes

**Requested Changes:**

I would recommend to highlight more explicitly in the introduction the fact that connections between message passing in GNNs and opinion dynamics have been explored before, so that the contribution of the paper is more accurately placed into the existing literature.

**Strengths And Weaknesses:**

Strengths: The paper is based on an interesting connection between neural message passing mechanisms in GNNs and opinion dynamics in social networks. To my knowledge, although it is a very natural connection, it has not been made explicit in this form before, and provides a number of new, interesting insights. In particular, this connection enables leveraging many insights developed in the line of work in opinion dynamics for a different problem, as the authors showcase extensively. I expect that more interesting results will follow from this connection in future work. The paper does a great job at providing sufficient background on key concepts in the literature on opinion dynamics, and how those translate in the context of message passing in GNNs, without assuming much familiarity with either area. They are also able to provide an intuitive and informative explanation regarding oversmoothing, a well-known problem in the literature. The main contribution of the paper is to introduce a new mechanism for message passing, which has a strong theoretical foundation as it is inspired by well-established work in opinion dynamics. The experimental results are overall very convincing and thorough, and demonstrate that the proposed mechanism outperforms prior approaches in the literature. The writing overall is of high quality, and it was a joy to read the paper. All prior work closely related to the paper appears to been discussed in detail. Moreover, all claims appear to be sound; I did not detect any notable issues. For those reasons, I believe merits acceptance.

In terms of weaknesses, the connection between message passing and opinion dynamics appears to have been studied before, albeit in a different form, as the authors discuss in the related work; in that sense, the novelty of the paper is somewhat limited. Further, from a technical standpoint, the results appear to be straightforward extensions of existing results that follow directly from the aforementioned connection. But I do not believe that the above caveats are a basis for rejection.

---

> ### Author Response · Authors · 2024-11-14
>
> Thank you for your positive feedback on our work and for your suggestion regarding additional discussion. Following your suggestion, we have added information in the introduction on previous explorations of the connections between message passing and opinion dynamics. On page 2, it now reads:
>
> > A few recent studies combine opinion dynamics to define propagation rules on graphs. However, they are based on position-based first-order message passing (Okawa & Iwata, 2022) or ignore data-driven propagation
> rules (Giráldez-Cru et al., 2022), resulting in limited expressivity.
>
> We hope that the revised version clearly highlights the novelty of our work.

---

### Review · Reviewer_h3KK · 2024-10-30

**Summary Of Contributions:**

This paper introduces ODNet, a novel neural message-passing framework for graph and hypergraph data inspired by sociological opinion dynamics models. ODNet applies concepts from these models to mitigate oversmoothing in Graph Neural Networks (GNNs) by incorporating bounded confidence into message propagation. The authors test the proposed model in both homophilic and heterophilic graphs and hypergraphs.

**Audience:**

Yes

**Broader Impact Concerns:**

No ethical issue.

**Claims And Evidence:**

Yes

**Requested Changes:**

1. How might ODNet’s mechanisms adapt to extremely large-scale graphs or real-time applications, and what optimizations could address potential scalability issues?
2. Would it be feasible to conduct a preliminary exploration of ODNet’s applicability to biological networks or other non-social domains to assess its broader utility?

3. Bellow I'm listing some missing citations:

*GNN*
[1] SCARSELLI, Franco, et al. The graph neural network model. IEEE transactions on neural networks, 2008, 20.1: 61-80.
[2] Kipf, Thomas N., and Max Welling. "Semi-supervised classification with graph convolutional networks." arXiv preprint arXiv:1609.02907 (2016).


*Opinion dynamics GNN related*
[1] Caralt FH, Gil GB, Duta I, Liò P, Cot EA. Joint Diffusion Processes as an Inductive Bias in Sheaf Neural Networks. arXiv preprint arXiv:2407.20597. 2024 Jul 30.
[2] Duta I, Cassarà G, Silvestri F, Liò P. Sheaf hypergraph networks. Advances in Neural Information Processing Systems. 2024 Feb 13;36

[3] Zaghen O, Longa A, Azzolin S, Telyatnikov L, Passerini A, Lio P. Sheaf Diffusion Goes Nonlinear: Enhancing GNNs with Adaptive Sheaf Laplacians. InICML 2024 Workshop on Geometry-grounded Representation Learning and Generative Modeling.
[4] Hansen, J. and Ghrist, R. Opinion dynamics on discourse sheaves. SIAM Journal on Applied Mathematics, 81(5): 2033–2060, 2021.

**Strengths And Weaknesses:**

**Strengths**

1. The application of opinion dynamics principles to neural message passing is creative and bridges distinct research fields, offering a fresh perspective on mitigating oversmoothing.
2. The paper provides solid theoretical support for ODNet’s mechanisms, alongside extensive experiments that demonstrate its competitive performance across multiple datasets.

**Weaknesses:**
1. Potential Generalizability Limitations: ODNet’s performance is validated on benchmark datasets, but its generalizability to other domains (e.g., biological networks) remains speculative.
2. While theoretically well-founded, the application of ODNet to very large graphs may raise scalability concerns that are not fully addressed in this paper.
3. Some citations are missing.

---

> ### Author Response · Authors · 2024-11-14
>
> Thank you for your positive feedback and constructive suggestions. We have revised the manuscript following your comments. Below is a point-by-point response to the concerns and requested changes.
>
> 1. **Scalability on Large Graphs**: The core design of ODNet centers on the potential term and bounded confidence. This design was specifically chosen to minimize additional computational costs. Bounded confidence is implemented through the influence function (Eq. 8 & 10) with a straightforward formulation. To demonstrate the scalability of our method to larger graphs, we included performance results on the ogb-arXiv dataset in Table 2 of the revision. The results of baseline methods are from GRAND. We did not validate the performance on larger graphs due to limited time for revision and, more importantly, because scalability is not the primary focus of this paper.
>
> 2. **Extension to Biological Networks**: In the revision, we provide a real-world case study demonstrating the effectiveness of ODNet in simplifying biological networks (Section 6.6). In this case study, we analyze and simplify microbiomes from two distinct environments, using samples collected from Mount Everest (ME) and Mount Trench (MT). In environmental microbiome analysis, examining the co-occurrence network of metabolic genes across different species can reveal potentially key biological genes that play dominant roles. However, the initial co-occurrence network can be very complex and contain many less important genes, making the simplification step important for an effective analysis. Although the ME and MT environments are distinctly different, they share common microbial species and metabolic genes. Environmental factors, however, significantly influence the co-occurrence network, and we expect distinct patterns in the simplified networks from each environment. We provide a detailed problem formulation, results analysis, and visualizations of the simplified networks in the main text. For additional information on the gene co-occurrence network and the simplification task, we included background information in Appendix C and basic statistics for the two network datasets in Appendix D.
>
> 3. **Missing Citations**: Thank you for recommending these references. We have added them to the revised manuscript.

---

### Comment · Action_Editor_QukE · 2024-10-30
**Reviews are in**

Dear authors, three reviews have been received. Please review and post any comments within the next two weeks.

---

### Decision · Action_Editor_QukE · 2024-12-02

**Recommendation:** Accept as is

**Comment:**

Reviewer comments have been sufficiently addressed during revision period.

**Audience:**

The reviewers all agreed that there is sufficient interest in this paper among the TMLR audience.

**Claims And Evidence:**

The reviewers agree that this paper makes contributions to connecting Neural message passing in GNNs and opinion dynamics in social networks. The reviewers found the results interesting and the paper well-written.